# UniPercept: Towards Unified Perceptual-Level Image Understanding across Aesthetics, Quality, Structure, and Texture

**Shuo Cao** [1 2 * +]   **Jiayang Li** [3 2 *]   **Xiaohui Li** [2 4]   **Yuandong Pu** [2 4]   **Kaiwen Zhu** [2 4]   **Yuanting Gao** [5]   **Siqi Luo** [2 4]
**Yi Xin** [2]   **Qi Qin** [2]   **Yu Zhou** [6]   **Xiangyu Chen** [7]   **Wenlong Zhang** [2]   **Bin Fu** [2]   **Yu Qiao** [2]   **Yihao Liu** [2 †]

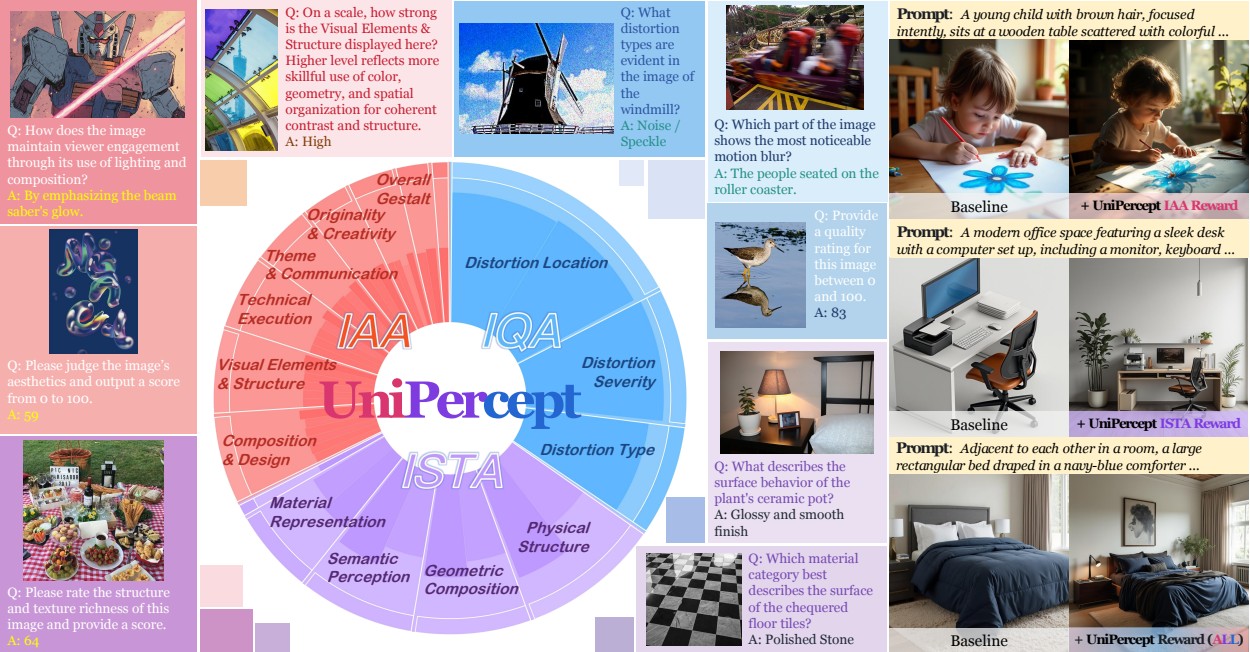

*Figure 1.* **Overview of UniPercept.** We propose a unified framework for perceptual-level image understanding. Left: **UniPercept-Bench**, a hierarchical benchmark covering Image Aesthetics (**IAA**), Quality (**IQA**), and Structure & Texture (**ISTA**) assessment via Visual Rating (**VR**) and Visual Question Answering (**VQA**). Right: Visual comparisons show that employing **UniPercept as a reward model** for post-training significantly elevates T2I generation quality, particularly in aesthetic appeal, structural fidelity, and fine-grained textures.

## Abstract

Multimodal large language models (MLLMs) have achieved remarkable progress in visual understanding tasks such as visual grounding, segmentation, and captioning. However, their ability to perceive **perceptual-level** image features remains limited. In this work, we present **UniPercept-Bench**, a unified framework for *perceptual-level image understanding* across

three key domains: **Aesthetics**, **Quality**, **Structure and Texture**. We establish a hierarchical definition system and construct large-scale datasets to evaluate perceptual-level image understanding. Based on this foundation, we develop a strong baseline **UniPercept** trained via Domain-Adaptive Pre-Training and Task-Aligned RL, enabling robust generalization across both **Visual Rating (VR)** and **Visual Question Answering (VQA)** tasks. UniPercept outperforms existing MLLMs on perceptual-level image understanding and can serve as a **plug-and-play reward model** for text-to-image generation. This work defines Perceptual-Level Image Understanding in the era of MLLMs, providing a foundation for its advancement through a comprehensive benchmark and a strong baseline. The project page is available at https://thunderbolt215.github.io/Unipercept-project/.

---

[*]Equal contribution [+] Work done during his internship at Shanghai Artificial Intelligence Laboratory.   [1]University of Science and Technology of China [2]Shanghai Artificial Intelligence Laboratory [3]Peking University [4]Shanghai Jiao Tong University [5]Tsinghua University [6]Sun Yat-sen University [7]Institute of Artificial Intelligence (TeleAI), China Telecom. Correspondence to: Yihao Liu <liuyihao@pjlab.org.cn>.

*Proceedings of the 43rd International Conference on Machine Learning*, Seoul, South Korea. PMLR 306, 2026. Copyright 2026 by the author(s).

# 1. Introduction

Recent years have witnessed the rapid advancement of multi-modal large language models (MLLMs), which now achieve impressive performance across a variety of vision–language tasks including segmentation, visual grounding, image captioning, and visual reasoning (Achiam et al., 2023; Team et al., 2023; Lindsey et al., 2025). These advancements are largely driven by their strong capability to learn and align semantic-level representations, allowing models to identify objects and scenes, capture their relationships and perform visual reasoning (Bai et al., 2025; Wang et al., 2025).

However, despite extensive progress on semantic understanding, the perceptual-level comprehension of images—how humans perceive aesthetics, quality, structure, and texture—remains substantially underexplored. As illustrated in Fig. 2, **semantic-level** tasks focus on high-level interpretation of visual entities (e.g., object attributes or contextual reasoning), whereas **perceptual-level** tasks require assessing fine-grained, low-level visual appearance, such as aesthetic harmony, degradation severity, structural regularity, or surface texture. These perceptual attributes are inherently subtle, often subjective, and closely tied to human visual experience, making them fundamentally different from typical semantic-level tasks.

Human visual perception involves much more than object recognition: it includes nuanced judgments about *how* an image looks and feels. Such perceptual cues play a crucial role in many downstream applications (e.g., content creation, image enhancement, and generative model alignment). Nevertheless, current MLLMs often struggle with these aspects, producing unstable or inconsistent predictions when evaluating aesthetic quality, perceptual degradation, or structural coherence. This gap highlights the need for a unified framework that can *explicitly define, evaluate, and improve* perceptual-level understanding in MLLMs, as perceptual attributes remain far less standardized and less explored than semantic ones. Addressing this missing layer is essential for building models that align more closely with human judgments, thereby achieving higher visual quality.

To bridge this gap, we propose **UniPercept**, the first unified framework for **perceptual-level image understanding** across three core domains: Image Aesthetics Assessment (**IAA**), Image Quality Assessment (**IQA**), and Image Structure and Texture Assessment (**ISTA**). Our contributions are summarized as follows:

**UniPercept-Bench.** As illustrated in Fig. 1 and further exemplified in Fig. 3, we establish a comprehensive hierarchical taxonomy of perceptual attributes, consisting of three progressive layers: *Domain–Category–Criterion*. Building upon this taxonomy, we construct UniPercept-Bench, a systematically designed benchmark for evaluating perceptual-

### Semantic-level Understanding

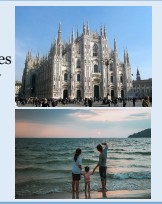

**Image Captioning**
*Q: Generate a natural-language description of the scene.*
A: The image shows a grand cathedral with intricate spires and stone carvings under a clear blue sky, surrounded by people in the open square.

**Object Visual Reasoning**
*Q: What are the people looking at near the shore?*
A: The sunset over the ocean.

*Interpreting Context & Meaning*

### Perceptual-level Understanding

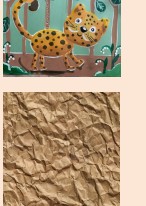

**Aesthetics Assessment**
*Q: Evaluate visual balance.*
A: The central placement of the cat is a key factor in achieving visual balance in this image. The cat acts as the focal point, drawing immediate attention and anchoring ...

**Structure Assessment**
*Q: Assess surface texture.*
A: The image depicts a surface with a crumpled texture, characterized by irregular folds, creases, and a matte finish. The characteristics ...

*Perceiving Appearance & Quality*

*Figure 2.* **Semantic-level** vs. **Perceptual-level** understanding.

level image understanding in MLLMs. The benchmark covers fine-grained perceptual attributes and supports both Visual Rating (**VR**) and Visual Question Answering (**VQA**) tasks for unified perceptual-level image understanding.

**UniPercept.** We develop **UniPercept** as a strong baseline MLLM through large-scale Domain-Adaptive Pre-Training and task-aligned reinforcement learning. Without relying on additional human feedback, the model learns to reliably assess perceptual attributes across diverse visual domains. UniPercept demonstrates strong generalization across both **VR** and **VQA** tasks, and achieves consistent gains in all three perceptual domains (**IAA**, **IQA**, and **ISTA**), substantially outperforming state-of-the-art MLLMs.

**Applications of UniPercept.** UniPercept serves as a strong plug-and-play **reward model** for post-training T2I models (Liu et al., 2025), enabling direct optimization of perceptual-level signals such as aesthetic quality, structural richness, and scene diversity, as shown in Fig. 1. This integration yields clear and controllable improvements in the perceptual quality of generated images. Beyond reward optimization, UniPercept serves as a unified metric for image evaluation and a tool for characterizing perceptual distributions in large-scale datasets.

# 2. Related Works

## 2.1. MLLM Benchmark

With the rapid development of MLLMs, evaluating their performance has gone far beyond simple semantic understanding tasks such as image recognition or segmentation. In recent years, researchers have begun to assess whether a model can achieve a deeper level of understanding and reasoning about visual content.

For instance, MMMU (Yue et al., 2024a) focuses on university-level exam questions across diverse disciplines, while MMMU-Pro (Yue et al., 2024b) extends this idea

*Figure 3.* **Representative QA examples in UniPercept-Bench.** Questions follow a three-level hierarchy of *Domain–Category–Criterion*, defining perceptual scope, specific visual aspects, and fine-grained criteria for constructing diverse, perception-oriented VQA tasks.

with more complex, cross-domain reasoning challenges. MEGA-Bench (Chen et al., 2024b) emphasizes large-scale multimodal comprehension and knowledge integration, and MMStar (Chen et al., 2024c) targets general reasoning and factual understanding in visual contexts. MMBench (Liu et al., 2024a) evaluates comprehensive perception and reasoning across everyday images, MathVista (Lu et al., 2023) centers on mathematical and geometric reasoning within visual scenes, and OCRBench (Liu et al., 2024b) specifically tests a model's capability to recognize and interpret text embedded in images.

However, these benchmarks rely on converting visual content into text representations before reasoning, emphasizing language-based inference over genuine visual understanding. In contrast, UniPercept-Bench directly evaluates perceptual-level visual properties—such as technical execution, distortion location, and material depiction—bridging the gap between perceptual and semantic understanding.

## 2.2. Image Assessment

For perceptual-level image assessment, prior research has primarily focused on two major areas: Image Aesthetics Assessment (IAA) and Image Quality Assessment (IQA). Extensive benchmarks and methods have been developed for these two categories—for example, Q-Align (Wu et al.,

2024a), UNIAA (Zhou et al., 2024), and ArtiMuse (Cao et al., 2025) for IAA, and MUSIQ (Ke et al., 2021), Depict-QA (You et al., 2024; 2025b), DeQA (You et al., 2025a), and Q-Insight (Li et al., 2025b) for IQA. In contrast, another crucial perceptual dimension—Image Structure and Texture Assessment (**ISTA**)—has received far less systematic attention. Although a few prior works (Cimpoi et al., 2014; Sharan et al., 2014) touch upon aspects of structural or textural perception, they do not provide a unified or comprehensive definition of ISTA, leaving this important component of perceptual-level understanding still insufficiently explored.

In addition, most existing datasets focus on a single aspect such as numerical scoring or question answering, without providing a comprehensive and multi-dimensional evaluation framework. As multimodal large models continue to improve, many existing models have already achieved very high accuracy on prior benchmarks (Huang et al., 2024; Wu et al., 2023a; Zhang et al., 2024), reducing the ability of these benchmarks to effectively distinguish stronger models. In contrast, our UniPercept-Bench addresses these limitations by covering multiple perceptual dimensions, offering diverse and detailed evaluation data, and presenting a more comprehensive challenge for MLLMs.

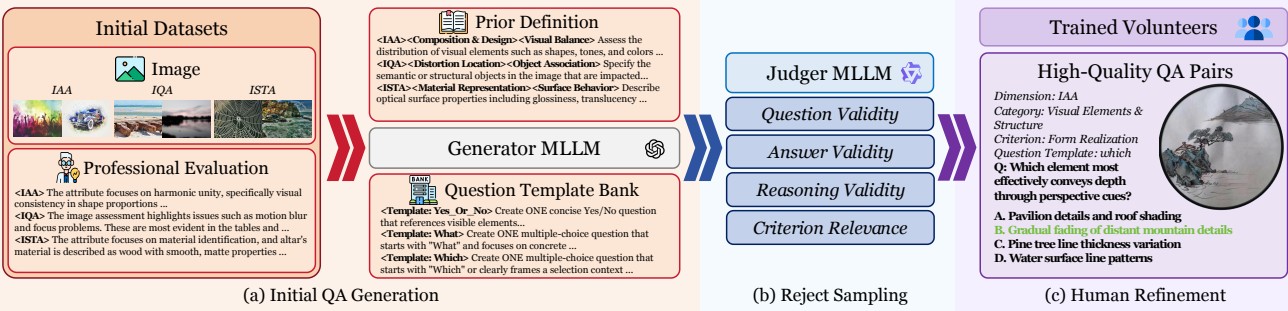

*Figure 4.* **Constuction pipeline of UniPercept-Bench.** A three-stage process initial QA generation, reject sampling, and human refinement to produce high-quality perceptual-level QA pairs across aesthetics, quality, structure, and texture.

## 3. UniPercept-Bench

### 3.1. Definition

As shown in Fig. 1, UniPercept-Bench targets **perceptual-level** image understanding across Image Aesthetics Assessment (IAA), Image Quality Assessment (IQA), and Image Structure and Texture Assessment (ISTA). **IAA** focuses on the *perceived aesthetic attributes* of an image, such as composition, style, emotion, and overall visual appeal. **IQA** targets the *perceived fidelity and degradation factors*, including noise, blur, compression artifacts, and overall distortion levels. **ISTA** evaluates the *structural and textural characteristics* of a scene, emphasizing geometry, material properties, and local detail richness. Although all three domains assess images, the tasks focus on fundamentally **different aspects**. For example, a high-quality image may not possess strong aesthetic value, while an aesthetically pleasing image may contain only simple or sparse textures.

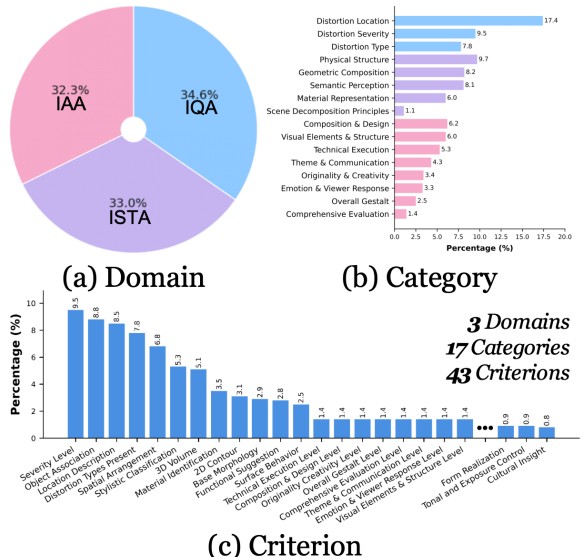

(a) Domain  (b) Category

**3 Domains**
**17 Categories**
**43 Criterions**

(c) Criterion

*Figure 5.* **Distribution of UniPercept-Bench** across (a) Domain, (b) Category, and (c) Criterion. Zoom in for best view.

While IAA and IQA have been widely studied in prior works such as Q-Align (Wu et al., 2024a), the DepictQA series (You et al., 2024; 2025b;a), and ArtiMuse (Cao et al.,

2025), **ISTA** remains largely unexplored, with little prior effort to systematically define or evaluate it. Inspired by recent advances in image generation and low-level vision (Labs et al., 2025; Wu et al., 2025a; Yu et al., 2024), we reorganize and unify the existing definition systems for IAA and IQA, and propose the **first** systematic, operational definition of ISTA. Together, these three domains form a coherent perceptual-level framework that enables comprehensive and fine-grained assessment of IAA, IQA ans ISTA cues aligned with human perception. More details are provided in the Appendix A, B, C.

### 3.2. Benchmark Construction

We organize UniPercept-Bench using a three-tier taxonomy of *Domain–Category–Criterion*. For IAA and IQA, we consolidate expert-agreed definitions from prior literature. For ISTA, we conduct structured interviews with domain practitioners to derive precise, consensus-driven perceptual criteria. Fig. 5 summarizes the overall distribution of UniPercept-Bench. Building on this taxonomy, UniPercept-Bench introduces two complementary task forms:

**Visual Rating (VR).** Models output a continuous score representing perceptual alignment across IAA (aesthetics quality) and IQA (image quality), serving as a quantitative measure of perceptual understanding. We additionally define the **ISTA Rating** task to evaluate *structure and texture richness*, where a higher score indicates a scene with more complex geometry and richer textures. Following the definition framework of UniPercept-Bench, we design a scoring method that computes the ISTA score by aggregating and weighting the counts of structured attributes extracted from each image, as detailed in the Appendix B.3.

**Visual Question Answering (VQA).** Question sets are constructed at the *Domain–Category–Criterion* levels to assess perceptual-level understanding and reasoning. Together, VR and VQA form a unified evaluation protocol that jointly measures **quantitative judgment** and **explanatory consistency**, advancing comprehensive perceptual-level understanding, as illustrated in Fig. 3.

**Initial QA Generation.** As illustrated in Fig. 4, We be-

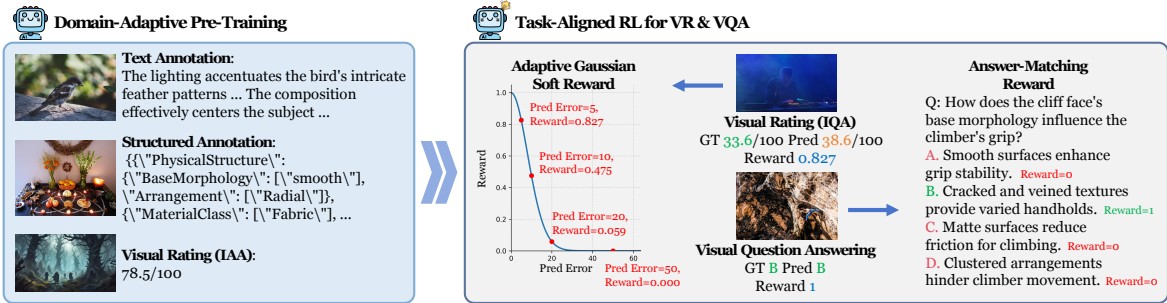

*Figure 6.* **Training pipeline of UniPercept.** A two-stage framework combining domain-adaptive pre-training for perceptual understanding and task-aligned RL to jointly optimize Visual Rating and Visual Question Answering.

gin by collecting images from three perceptual dimensions to ensure diversity across aesthetic quality, distortion severity, and structural complexity. For IAA and IQA, we adopt expert-annotated datasets (Cao et al., 2025; You et al., 2025b; Chen et al., 2024a) as professional evaluations. For ISTA, we introduce structured annotations guided by UniPercept-Bench's taxonomy, describing scene composition, geometric layout, and material representation. These annotations are paired with predefined *Domain–Category–Criterion* definitions and matched templates from a curated *Question Template Bank*. The *Generator MLLM* (GPT-4o (Achiam et al., 2023)) combines the image, annotation, and template to produce candidate QA pairs along with brief reasoning rationales.

**Reject Sampling.** A heterogeneous *Judger MLLM* (Qwen-2.5-VL-78B-Instruct (Bai et al., 2025)) evaluates each QA pair on **Question Validity**, **Answer Validity**, **Reasoning Validity**, and **Criterion Relevance** using a five-point scale. Only samples rated *good* or above on all aspects are retained, removing about 40% of candidates.

**Human Refinement.** Finally, trained volunteers with expertise in Image Assessment conduct manual validation to ensure alignment with human reasoning and perception. Borderline cases are revised for clarity and consistency, while invalid samples are removed. The resulting dataset consists of high-quality QA pairs that are both perceptually grounded and semantically coherent.

# 4. UniPercept

## 4.1. Domain-Adaptive Pre-Training

We aim to train a model with perceptual-level understanding ability across Aesthetics, Quality, Structure, and Texture. To endow the model with a foundational capability for perceptual-level image understanding, we first conduct a **Domain-Adaptive Pre-Training** stage. Specifically, we select and process large-scale existing datasets, resulting in two types of pre-training data as follows:

**Text-based QA pairs.** We construct large-scale QA pairs for IAA, IQA, and ISTA. For ISTA, we additionally design

structured-output QA pairs (Fig. 6) to support fine-grained structural reasoning.

**VR-based QA pairs.** We additionally incorporate VR data for IAA, IQA, and ISTA, also visualized in Fig. 6, which allows the model to directly associate perceptual-level attributes with quantitative ratings.

In total, the dataset in domain-adaptive pre-training contains approximately **800K** samples. Through Domain-Adaptive Pre-Training, the model acquires the essential capability to handle diverse perceptual-level image understanding tasks across different domains.

## 4.2. Task-Aligned RL for VR & VQA

To achieve precise alignment of the model across both Visual Rating (VR) and Visual Question Answering (VQA), we employ the GRPO algorithm (Shao et al., 2024) to perform **Task-Aligned RL** with task-specific reward functions for VR and VQA. For the VQA task, we adopt a binary reward:

$$
r_{vqa} = \begin{cases} 1, & \text{if the predicted answer is correct,} \\ 0, & \text{otherwise.} \end{cases} \quad (1)
$$

For the VR task, we design an **Adaptive Gaussian Soft Reward**, which continuously evaluates the prediction according to its numerical deviation from the ground truth:

$$
r_{vr} = \exp\left(-\frac{(|p_i - g_i|)^2}{2\sigma_{\mathrm{dyn}}^2}\right), \sigma_{\mathrm{dyn}} = \sigma_0 \left(1 + \alpha\frac{|p_i - g_i|}{100}\right), \quad (2)
$$

where $p_i$ and $g_i$ denote the predicted and ground-truth scores (mapped to $[0, 100]$), $\sigma_0$ is the base smoothing coefficient, and $\alpha$ controls the degree of adaptive Gaussian smoothing. This soft reward offers smoother gradients and avoids threshold-induced discontinuities. Following prior works (Wu et al., 2024a; Cao et al., 2025), we adopt a *Token As Score* strategy for VR, deriving ratings from the predicted token distribution. We then incorporate this task-specific reward into the GRPO objective to enable perceptual-level policy optimization:

*Table 1.* **Performance comparison of different models on UniPercept-Bench-VR.** The best results are highlighted with dark blue cells, and the second-best results with light blue cells. Models with * indicate those retrained on ArtiMuse-10K, KonIQ-10K, and ISTA-10K. "-/-" denotes refusal or failure to respond. Results are reported as SRCC/PLCC.

| Models | VR - IAA | | | | | VR - IQA | | | | | VR - ISTA |
|---|---|---|---|---|---|---|---|---|---|---|---|
| | ArtiMuse-10K | AVA | TAD66K | FLICKR-AES | Avg. | KonIQ-10K | SPAQ | KADID | PIPAL | Avg. | ISTA-10K |
| *Proprietary Models* | | | | | *Large-scale Pretraining, Cross domain testing* | | | | | | |
| GPT-4o | 0.333/0.276 | 0.509/0.485 | 0.278/0.282 | 0.605/0.597 | 0.431/0.410 | 0.695/0.744 | 0.874/0.881 | 0.677/0.646 | 0.325/0.349 | 0.643/0.655 | -0.003/0.116 |
| Llama-4-Scout | 0.204/0.147 | 0.345/0.329 | 0.236/0.210 | 0.548/0.506 | 0.333/0.298 | 0.503/0.653 | -0.041/-0.007 | -0.099/-0.004 | -0.007/0.023 | 0.089/0.170 | -0.025/0.047 |
| Gemini-2.5-pro | 0.187/0.035 | 0.248/0.100 | 0.143/0.037 | 0.357/0.206 | 0.234/0.095 | 0.582/0.316 | 0.087/0.212 | 0.436/0.274 | 0.225/-0.019 | 0.333/0.196 | -0.230/-0.118 |
| Claude-Sonnet-4.5 | 0.041/0.027 | 0.003/0.013 | 0.040/0.047 | 0.037/0.049 | 0.030/0.034 | -0.037/-0.043 | 0.036/0.085 | 0.223/0.273 | -0.131/-0.088 | 0.023/0.057 | 0.125/0.089 |
| Claude-Sonnet-4.5-Think | 0.066/0.103 | 0.018/0.019 | 0.026/0.039 | -/- | 0.037/0.054 | -/- | -/- | -/- | -/- | -/- | -/- |
| *Open-Source Models* | | | | | *Large-scale Pretraining, Cross domain testing* | | | | | | |
| LLaVA-OneVision-1.5-Instruct-8B | 0.274/0.212 | 0.381/0.378 | 0.213/0.224 | 0.586/0.541 | 0.364/0.339 | 0.639/0.744 | 0.505/0.534 | 0.417/0.407 | | 0.520/0.562 | -0.094/0.027 |
| GLM-4.5-V-106BA12B | 0.346/0.249 | 0.464/0.420 | 0.289/0.278 | 0.651/0.597 | 0.438/0.386 | 0.721/0.765 | -0.040/-0.038 | -0.142/-0.128 | 0.013/0.020 | 0.138/0.155 | 0.083/0.117 |
| InternVL3-8B | 0.245/0.211 | 0.372/0.344 | 0.205/0.191 | 0.547/0.476 | 0.342/0.306 | 0.574/0.646 | 0.828/0.800 | 0.496/0.475 | 0.435/0.459 | 0.583/0.595 | -0.127/0.046 |
| InternVL3-78B | 0.223/0.206 | 0.385/0.344 | 0.221/0.220 | 0.518/0.433 | 0.337/0.301 | 0.635/0.676 | 0.849/0.852 | 0.579/0.553 | 0.415/0.457 | 0.619/0.634 | -/- |
| InternVL3.5-8B | 0.135/0.104 | 0.308/0.295 | 0.180/0.182 | 0.519/0.448 | 0.286/0.257 | 0.663/0.660 | 0.783/0.777 | 0.541/0.478 | 0.351/0.386 | 0.585/0.575 | -0.096/-0.025 |
| InternVL3.5-38B | 0.219/0.175 | 0.359/0.357 | 0.201/0.208 | 0.559/0.529 | 0.334/0.317 | 0.578/0.652 | 0.840/0.831 | 0.568/0.537 | 0.448/0.457 | 0.608/0.619 | 0.262/0.345 |
| QwenVL-2.5-Instruct-7B | 0.223/0.143 | 0.359/0.324 | 0.208/0.195 | 0.588/0.520 | 0.345/0.296 | 0.708/0.762 | -/- | 0.521/0.517 | 0.350/0.361 | 0.526/0.547 | -0.046/0.076 |
| QwenVL-2.5-Instruct-72B | 0.233/0.197 | 0.408/0.387 | 0.232/0.235 | 0.626/0.589 | 0.375/0.352 | 0.762/0.820 | -/- | 0.606/0.570 | 0.381/0.407 | 0.583/0.599 | 0.091/0.148 |
| QwenVL-3-Instruct-8B | 0.156/0.094 | 0.280/0.170 | 0.191/0.121 | 0.507/0.388 | 0.283/0.193 | 0.761/0.822 | 0.612/0.604 | 0.723/0.696 | 0.434/0.427 | 0.633/0.637 | 0.033/0.044 |
| QwenVL-3-Instruct-32B | 0.227/0.130 | 0.353/0.198 | | 0.200/0.095 | 0.338/0.209 | 0.796/0.838 | 0.690/0.657 | 0.673/0.682 | 0.414/0.402 | 0.673/0.644 | 0.084/0.106 |
| *Specialized Models* | *In domain* | *Cross domain* | | | *Avg.* | *In domain* | *Cross domain* | | | *Avg.* | *In domain* |
| ArtiMuse | 0.614/0.627 | 0.397/0.385 | 0.230/0.232 | 0.349/0.334 | 0.398/0.395 | -/- | -/- | -/- | -/- | -/- | -/- |
| DeQA | -/- | -/- | -/- | -/- | -/- | 0.953/0.941 | 0.895/0.896 | 0.694/0.687 | 0.472/0.478 | 0.753/0.750 | -/- |
| Q-Align* | 0.551/0.573 | 0.398/0.386 | 0.194/0.197 | 0.137/0.123 | 0.320/0.320 | 0.941/0.940 | 0.886/0.887 | 0.674/0.684 | 0.403/0.419 | 0.726/0.733 | -/- |
| Q-Insight | -/- | -/- | -/- | -/- | -/- | 0.933/0.916 | 0.907/0.905 | 0.742/0.736 | 0.486/0.474 | 0.767/0.758 | -/- |
| Q-Insight* | 0.228/0.175 | 0.405/0.376 | 0.212/0.217 | 0.617/0.537 | 0.366/0.326 | 0.733/0.750 | 0.800/0.938 | 0.580/0.548 | 0.369/0.368 | 0.621/0.651 | 0.060/0.152 |
| **UniPercept (Ours)** | **0.746/0.738** | **0.589/0.577** | **0.336/0.346** | **0.688/0.681** | **0.590/0.586** | **0.940/0.949** | **0.904/0.895** | **0.872/0.870** | **0.581/0.594** | **0.824/0.827** | **0.778/0.767** |

$$\mathcal{J}_{\text{GRPO}}^{\mathcal{B}}(\theta) = \mathbb{E}_{\mathcal{B}}\left[\frac{1}{\sum_{i=1}^{G}|o^i|}\sum_{i=1}^{G}\sum_{t=1}^{|o^i|}r_i \cdot \min\Big(r_t^i(\theta)\hat{A}_t^i,\right.$$

$$\left.\text{clip}\big(r_t^i(\theta), 1-\epsilon, 1+\epsilon\big)\hat{A}_t^i\Big)\right]. \tag{3}$$

where $r_t^i(\theta) = \frac{\pi_\theta(o_t^i|q^i, o_{<t}^i)}{\pi_{\text{old}}(o_t^i|q^i, o_{<t}^i)}$ is the ratio between current and old policies, $\hat{A}_t^i$ is the estimated advantage, and $r_i$ is the task-specific reward from Eq. 1 or Eq. 2. This unified formulation enables GRPO to align model behavior with both discrete correctness (VQA) and continuous perceptual consistency (VR).

# 5. Experiments

## 5.1. Implementation

**Evaluated Models.** We evaluate a total of 18 models, encompassing three categories. (1) Proprietary Models: GPT-4o (Achiam et al., 2023), Llama-4-Scout (AI, 2025), Gemini-2.5-Pro (Team et al., 2023), Claude-Sonnet-4.5 (Lindsey et al., 2025) and Claude-Sonnet-4.5-Think (Lindsey et al., 2025). (2) Leading Open-Source Models: the InternVL3 and InternVL3.5 series (Zhu et al., 2025; Wang et al., 2025), QwenVL-2.5-Instruct and QwenVL-3-Instruct series (Bai et al., 2025), GLM-4.5-V-106BA12B (Team et al., 2025), as well as LLaVA-OneVision-1.5-Instruct (An et al., 2025). (3) Specialized Models for IAA and IQA: Q-Align (Wu et al., 2024a), ArtiMuse (Cao et al., 2025), DeQA (You et al., 2025a), and Q-Insight (Li et al., 2025b).

**Evaluation Settings.** For VQA, all models were provided with identical prompts corresponding to each question, and their generated answers were compared against the ground-truth options. For VR, we designed task-specific prompts for models lacking a dedicated scoring interface to elicit quantitative predictions, while specialized models for visual rating were directly evaluated through their native interfaces. Specifically, VR evaluation covers three sub-tasks: (i) IAA (ArtiMuse-10K (Cao et al., 2025), AVA (Murray et al., 2012), TAD66K (He et al., 2022), FLICKR-AES (Ren et al., 2017)), (ii) IQA (KonIQ-10K (Hosu et al., 2020), SPAQ (Fang et al., 2020), KADID (Lin et al., 2019), PIPAL (Gu et al., 2020)), and (iii) ISTA (ISTA-10K). It is worth noting that all specialized models are trained exclusively on *In-domain* datasets. For representative models (Q-Align and Q-Insight), we further retrain them on a mixed dataset comprising ArtiMuse-10K, KonIQ-10K, and ISTA-10K to ensure consistent perceptual alignment.

**Training Details.** Based on InternVL3-8B(Zhu et al., 2025), UniPercept is trained in two stages as described in Sec. 4. The Domain-Adaptive Pre-Training stage adopts multiple public datasets, including APDDv2 (Jin et al., 2024) and Impressions (Kruk et al., 2023) for IAA, Q-Ground-100K (Chen et al., 2024a) and DataDepictQA (You et al., 2025b) for IQA, and DTD (Cimpoi et al., 2014), FMD (Sharan et al., 2014), Flickr2K (Lim et al., 2017), and LSDIR (Li et al., 2023) for ISTA, among others. After preprocessing and filtering, the resulting corpus contains approximately 800K samples in total. For Task-Aligned RL, we adopt the VR datasets ArtiMuse-10K (Cao et al., 2025), KonIQ-10K (Hosu et al., 2020), and ISTA-10K, along with ~30K VQA samples generated as described in Sec. 3.2. Training takes 2 epochs per stage on 16 NVIDIA A100 GPUs with a batch size of 128. In GRPO, we sample $n=8$ responses per query and set $\beta=0.001$, $\varepsilon=0.2$, and $\sigma=0.8$.

## 5.2. Benchmark Results with Analysis

### 5.2.1. VISUAL RATING

**General Models vs. Specialized Models.** Visual Rating is a challenging task that requires models to output continuous, high-precision perceptual scores. As shown in Table 1,

*Table 2.* **Performance comparison of different models on UniPercept-Bench-VQA (IAA).** Category names are abbreviated, with full definitions provided in the Appendix C.3. Results follow the same notation throughout the paper.

| Models | IAA Categories | | | | | | | | QA Templates | | | | | | Overall |
|---|---|---|---|---|---|---|---|---|---|---|---|---|---|---|---|
| | Comp. | VisStr. | Tech. | Creat. | Theme. | Emo. | Gest. | CompEv. | Lv.Pred | How | What | Which | Why | Yes-No | |
| *Random Guess* | 23.08% | 27.27% | 21.95% | 29.63% | 25.93% | 22.86% | 23.68% | 32.56% | 24.14% | 21.28% | 30.43% | 25.32% | 24.00% | 29.49% | 25.17% |
| *Proprietary Models* | | | | | | | | | | | | | | | |
| GPT-4o | 64.62% | 59.57% | 57.58% | 60.19% | 65.19% | 67.62% | 51.95% | 30.23% | 38.86% | 78.17% | 72.46% | 62.66% | 72.67% | 70.51% | 60.04% |
| Llama-4-Scout | 62.56% | 68.45% | 59.76% | 61.11% | 57.78% | 70.48% | 48.68% | 32.56% | 43.97% | 70.92% | 69.57% | 61.39% | 77.33% | 70.51% | 60.91% |
| Gemini-2.5-pro | 71.79% | 68.45% | 61.59% | 76.85% | 67.41% | 63.81% | 61.84% | 37.21% | 45.98% | 78.72% | 73.91% | 67.72% | 84.67% | 84.62% | 66.44% |
| Claude-Sonnet-4.5 | 70.26% | 70.05% | 62.20% | 71.30% | 62.96% | 67.62% | 50.00% | 46.51% | 44.57% | 77.30% | 76.09% | 65.19% | 86.00% | 69.23% | 65.45% |
| Claude-Sonnet-4.5-Think | 71.28% | 69.52% | 61.21% | 68.52% | 62.22% | 66.67% | 53.25% | 41.86% | 44.57% | 75.89% | 77.54% | 67.09% | 86.00% | 66.67% | 64.73% |
| *Open-Source Models* | | | | | | | | | | | | | | | |
| LLaVA-OneVision-1.5-Instruct-8B | 67.18% | 68.62% | 61.21% | 62.96% | 67.41% | 62.86% | 53.25% | 20.93% | 34.86% | 85.21% | 79.71% | 65.82% | 83.33% | 69.23% | 62.60% |
| GLM-4.5-V-106BA12B | 67.18% | 65.78% | 60.98% | 75.00% | 64.44% | 68.57% | 51.32% | 46.51% | 45.40% | 71.63% | 78.26% | 65.82% | 84.67% | 70.51% | 64.46% |
| InternVL3-8B | 65.64% | 67.55% | 59.39% | 67.59% | 69.63% | 62.86% | 50.65% | 25.58% | 36.00% | 81.69% | 71.01% | 67.72% | 86.00% | 71.79% | 62.60% |
| InternVL3-78B | 71.79% | 73.26% | 61.21% | 73.15% | 74.81% | 74.29% | 53.25% | 37.21% | 45.14% | 85.82% | 81.16% | 72.15% | 86.00% | 75.64% | 68.28% |
| InternVL3.5-8B | 32.31% | 29.41% | 30.30% | 26.85% | 28.89% | 26.67% | 23.38% | 9.30% | 17.14% | 41.13% | 26.81% | 19.62% | 36.00% | 58.97% | 28.18% |
| InternVL3.5-38B | 37.44% | 40.11% | 27.88% | 39.81% | 34.81% | 38.10% | 45.45% | 6.98% | 34.00% | 47.52% | 26.09% | 28.48% | 37.33% | 50.00% | 35.67% |
| QwenVL-2.5-Instruct-7B | 67.18% | 70.74% | 56.36% | 66.67% | 68.89% | 63.81% | 48.05% | 37.21% | 38.86% | 76.76% | 75.36% | 67.09% | 87.33% | 71.79% | 63.19% |
| QwenVL-2.5-Instruct-72B | 22.05% | 24.60% | 25.45% | 29.63% | 30.37% | 18.10% | 19.48% | 6.98% | 14.00% | 19.86% | 17.39% | 24.05% | 41.33% | 51.28% | 23.74% |
| QwenVL-3-Instruct-8B | 31.28% | 32.09% | 32.12% | 37.04% | 34.07% | 22.86% | 37.66% | 25.58% | 35.43% | 14.89% | 17.39% | 34.81% | 26.67% | 73.08% | 31.92% |
| QwenVL-3-Instruct-32B | 23.08% | 26.74% | 32.12% | 26.85% | 32.59% | 20.95% | 33.77% | 20.93% | 33.43% | 9.22% | 13.77% | 31.01% | 18.67% | 66.67% | 27.39% |
| *Specialized Models* | | | | | | | | | | | | | | | |
| ArtiMuse | 67.69% | 68.45% | 64.85% | 74.07% | 71.85% | 64.76% | 61.04% | 32.56% | 39.14% | 88.65% | 76.81% | 72.78% | 85.33% | 79.49% | 66.31% |
| **UniPercept (Ours)** | **80.00%** | **77.54%** | **69.70%** | **80.56%** | **79.26%** | **80.95%** | **67.53%** | **69.77%** | **63.71%** | **92.20%** | **81.88%** | **75.32%** | **86.67%** | **84.62%** | **76.55%** |

*Table 3.* **Performance comparison of different models on UniPercept-Bench-VQA (IQA).**

| Models | IQA Categories | | | QA Templates | | | | | | Overall |
|---|---|---|---|---|---|---|---|---|---|---|
| | Loc. | Sev. | Type. | Lv.Pred | How | What | Which | Why | Yes-No | |
| *Random Guess* | 23.67% | 24.75% | 20.08% | 24.75% | 27.03% | 16.05% | 25.00% | 21.39% | 22.99% | 23.16% |
| *Proprietary Models* | | | | | | | | | | |
| GPT-4o | 71.74% | 53.18% | 70.49% | 53.18% | 83.78% | 59.26% | 61.31% | 80.21% | 67.82% | 66.36% |
| Llama-4-Scout | 60.18% | 58.19% | 52.05% | 58.19% | 82.16% | 37.04% | 38.69% | 66.31% | 62.07% | 57.81% |
| Gemini-2.5-pro | 32.84% | 52.84% | 40.98% | 52.84% | 40.54% | 32.72% | 29.17% | 41.18% | 28.74% | 40.17% |
| Claude-Sonnet-4.5 | 71.19% | 51.51% | 66.80% | 51.51% | 90.81% | 50.00% | 50.60% | 82.89% | 71.26% | 64.80% |
| Claude-Sonnet-4.5-Think | 71.19% | 55.52% | 66.80% | 55.52% | 89.19% | 50.00% | 51.79% | 82.89% | 72.41% | 65.90% |
| *Open-Source Models* | | | | | | | | | | |
| LLaVA-OneVision-1.5-Instruct-8B | 76.51% | 59.87% | 77.46% | 59.87% | 91.35% | 70.37% | 61.31% | 82.35% | 75.86% | 72.15% |
| GLM-4.5-V-106BA12B | 70.09% | 35.79% | 54.51% | 35.79% | 88.11% | 48.77% | 44.05% | 74.33% | 68.97% | 57.17% |
| InternVL3-8B | 71.56% | 52.84% | 59.43% | 52.84% | 87.03% | 59.88% | 48.81% | 71.12% | 71.26% | 63.69% |
| InternVL3-78B | 75.41% | 51.84% | 81.56% | 51.84% | 93.51% | 66.67% | 63.10% | 88.24% | 66.67% | 70.31% |
| InternVL3.5-8B | 38.17% | 44.82% | 38.11% | 44.82% | 35.14% | 41.98% | 30.36% | 36.36% | 56.32% | 39.98% |
| InternVL3.5-38B | 38.90% | 49.83% | 45.08% | 49.83% | 46.49% | 41.36% | 31.55% | 33.16% | 62.07% | 43.29% |
| QwenVL-2.5-Instruct-7B | 74.13% | 48.83% | 66.39% | 48.83% | 88.65% | 60.49% | 53.57% | 78.61% | 77.01% | 65.44% |
| QwenVL-2.5-Instruct-72B | 31.01% | 4.68% | 16.39% | 4.68% | 35.14% | 14.81% | 11.31% | 22.99% | 66.67% | 20.50% |
| QwenVL-3-Instruct-8B | 34.68% | 55.18% | 16.39% | 55.18% | 29.20% | 18.52% | 27.38% | 25.67% | 77.01% | 36.21% |
| QwenVL-3-Instruct-32B | 29.54% | 14.38% | 16.80% | 14.38% | 11.89% | 18.52% | 25.60% | 22.46% | 74.71% | 22.52% |
| **UniPercept (Ours)** | **77.43%** | **79.60%** | **90.98%** | **79.60%** | **87.03%** | **80.86%** | **75.60%** | **83.42%** | **79.31%** | **81.07%** |

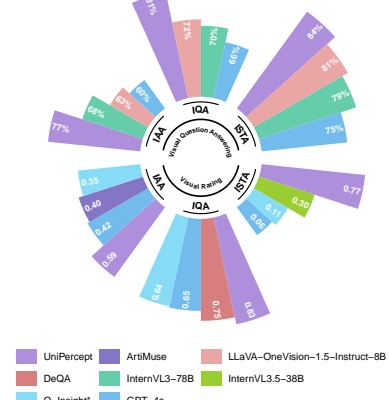

*Figure 7.* **Results on UniPercept-Bench.**

*Table 4.* **Performance comparison of different models on UniPercept-Bench-VQA (ISTA).**

| Models | ISTA Categories | | | | | QA Templates | | | | | Overall |
|---|---|---|---|---|---|---|---|---|---|---|---|
| | Scene. | Phys. | Mat. | Geo. | Sem. | How | What | Which | Why | Yes-No | |
| *Random Guess* | 26.50% | 23.63% | 24.73% | 30.30% | 30.58% | 26.28% | 23.84% | 24.29% | 33.77% | 33.33% | 26.60% |
| *Proprietary Models* | | | | | | | | | | | |
| GPT-4o | 75.64% | 79.12% | 73.48% | 33.33% | 77.27% | 71.79% | 78.78% | 69.23% | 77.92% | 72.46% | 74.64% |
| Llama-4-Scout | 73.50% | 75.27% | 71.68% | 72.73% | 67.77% | 75.64% | 69.77% | 69.64% | 77.27% | 69.57% | 71.86% |
| Gemini-2.5-pro | 76.50% | 82.42% | 77.06% | 66.67% | 77.69% | 78.21% | 78.20% | 75.71% | 82.47% | 71.01% | 77.73% |
| Claude-Sonnet-4.5 | 76.92% | 78.57% | 74.91% | 90.91% | 77.69% | 76.92% | 77.03% | 74.49% | 81.82% | 79.71% | 77.32% |
| Claude-Sonnet-4.5-Think | 77.35% | 78.02% | 73.12% | 87.88% | 75.21% | 76.28% | 74.71% | 74.09% | 81.82% | 76.81% | 76.08% |
| *Open-Source Models* | | | | | | | | | | | |
| LLaVA-OneVision-1.5-Instruct-8B | 78.63% | 85.16% | 82.44% | 72.73% | 80.17% | 83.33% | 81.40% | 75.30% | 84.42% | 88.41% | 81.13% |
| GLM-4.5-V-106BA12B | 81.20% | 79.67% | 74.55% | 72.73% | 75.21% | 80.77% | 76.74% | 73.68% | 79.87% | 78.26% | 77.22% |
| InternVL3-8B | 75.64% | 79.12% | 73.48% | 33.33% | 77.27% | 71.79% | 78.78% | 69.23% | 77.92% | 72.46% | 74.64% |
| InternVL3-78B | 79.06% | 85.16% | 77.42% | 69.70% | 78.51% | 81.41% | 79.65% | 73.68% | 84.42% | 81.16% | 79.28% |
| InternVL3.5-8B | 54.27% | 50.55% | 58.42% | 39.39% | 36.36% | 46.79% | 56.69% | 48.58% | 29.87% | 71.01% | 49.79% |
| InternVL3.5-38B | 50.00% | 55.49% | 61.29% | 30.30% | 35.95% | 50.64% | 59.30% | 42.91% | 37.01% | 57.97% | 50.10% |
| QwenVL-2.5-Instruct-7B | 74.79% | 72.53% | 74.91% | 51.52% | 73.55% | 73.72% | 77.33% | 66.80% | 74.03% | 73.91% | 73.30% |
| QwenVL-2.5-Instruct-72B | 14.10% | 29.12% | 19.71% | 12.12% | 18.60% | 20.51% | 12.21% | 14.57% | 31.17% | 46.38% | 19.59% |
| QwenVL-3-Instruct-8B | 27.78% | 32.42% | 25.45% | 39.39% | 24.79% | 14.74% | 23.26% | 28.34% | 25.32% | 81.16% | 27.63% |
| QwenVL-3-Instruct-32B | 26.50% | 24.73% | 19.00% | 15.15% | 18.60% | 11.54% | 18.31% | 22.67% | 17.53% | 66.67% | 21.65% |
| **UniPercept (Ours)** | **89.74%** | **85.71%** | **82.44%** | **93.94%** | **78.51%** | **82.69%** | **89.24%** | **78.54%** | **83.12%** | **85.51%** | **84.23%** |

most *general-purpose MLLMs* without task-specific training exhibit significantly lower performance compared with *specialized models*. This gap arises from the inherent difficulty of directly generating numerical outputs in text form, which often leads to hallucinated or unstable predictions—a phenomenon also discussed in prior works (Wu et al., 2024a; Li et al., 2025a; Cao et al., 2025). In contrast, specialized methods adopt structured strategies such as *Token As Score*, effectively stabilizing the regression behavior of large models on continuous scales. (Wu et al., 2024a; Cao et al., 2025)

**Domain-wise Analysis.** Model performance varies significantly across different domains, reflecting differences in task objectivity. IQA achieves the highest correlations, followed by ISTA, while IAA remains the most challenging due to its subjective and ambiguous aesthetic judgments. These results suggest that perceptual-level visual rating becomes increasingly difficult as the task shifts from objective assessment toward subjective reasoning.

### 5.2.2. VISUAL QUESTION ANSWERING

**Domain-wise Analysis.** Across domains, the best-performing generalized models can reach accuracies of up to 68.28%, 72.15%, and 81.13% on IAA, IQA, and ISTA, respectively. However, their average accuracies drop to only **51.62%**, **51.45%**, and **60.56%**. This disparity suggests that perceptual-level visual question answering remains highly challenging for current MLLMs, as they often rely on unstable domain-specific heuristics rather than robust perceptual reasoning. Among the three domains, ISTA appears slightly easier, as it centers on objective physical properties (e.g., geometry, material) that align better with pretraining priors. In contrast, IAA and IQA require subjective reasoning over aesthetics and perceptual quality—nuanced judgments that general models are not explicitly optimized for.

**Category-wise Analysis.** Models generally perform better on holistic perception categories such as *Composition & Design* and *Theme & Communication*, which focus on the overall aesthetic and semantic coherence of an image. Most models achieve accuracies above **60%** in these categories, indicating that global scene perception is relatively well captured by vision–language pretraining. In contrast, performance drops substantially on fine-grained perceptual categories such as *Overall Gestalt*, *Material Representation*, and *Geometric Composition*, where most models remain below **50%**. These results suggest that while high-level holistic understanding is well learned, precise reasoning over local structures cues remains a key limitation.

**QA Template Analysis.** The *Level Prediction* questions require models to provide fine-grained evaluations of images along specific dimensions (as shown in Fig. 3). Due to the demand for precise quantitative reasoning, most MLLMs struggle with this type, achieving only around **36%** and **45%** average accuracy on the IAA and IQA domains, respectively. For other QA templates, "*Yes-No*" and "*Why*" questions—closer to higher-level reasoning and causal inference—allow generalized MLLMs to reach average accuracies above **60%**. In contrast, "*What*" and "*Which*" questions, which require detailed and localized visual analysis, remain challenging, reflecting the current models' limited capacity for fine-grained perceptual understanding.

### 5.3. Further Discussion on UniPercept

#### 5.3.1. PERFORMANCE

As shown in Fig. 7 and Tabs. 1, 2, 3, and 4, UniPercept is capable of handling both major types of perceptual-level image understanding tasks (*VR & VQA*) across the three perceptual domains of *IAA*, *IQA*, and *ISTA*. Specifically, VR and VQA tasks in Fig. 7 are evaluated using Acc. and $(SRCC+PLCC)/2$, respectively. Benefiting from the proposed **Domain-Adaptive Pre-Training** and **Task-Aligned RL** strategies, UniPercept consistently outperforms both generalized and specialized models. It not only achieves strong in-domain results but also exhibits remarkable cross-domain generalization, demonstrating its capability as a unified perceptual understanding baseline.

#### 5.3.2. APPLICATION

The perceptual-level understanding capability of UniPercept enables a broad range of downstream applications. A primary use case is employing its multi-dimensional visual rating ability—spanning *Aesthetics Quality* (IAA), *Image Quality* (IQA), and *Structure & Texture Richness* (ISTA)—as a **reward model** for text-to-image generation. We integrate all these ratings as reward into the Flow-GRPO (Liu et al., 2025) fine-tuning pipeline based on FLUX.1-dev (Labs et al., 2025). The results in Tab. 16 and Fig. 8 show that incorporating UniPercept reward leads to clear improvements in human preference (PickScore (Kirstain et al., 2023), HPSv3 (Ma et al., 2025)), quality (DeQA (You et al., 2025a)), and aesthetics metrics (LAION-Aes (Schuhmann & Beaumont, 2022), ArtiMuse (Cao et al., 2025)), substantially enhancing the perceptual quality of the generated images. UniPercept can also serve as a unified set of **perceptual metrics** for evaluating generated images, as reflected in the quantitative results in Tab. 16. More applications and details are provided in Appendix D.4, D.5 and F.

*Table 5.* **Performance of FLUX.1-dev w/ UniPercept Reward.**

| Models | Preference Score | | Image Quality | Image Aesthetics | | UniPercept Score | | |
|---|---|---|---|---|---|---|---|---|
| | PickScore | HPSv3 | DeQA | LAION-Aes | ArtiMuse | IAA | IQA | ISTA |
| Baseline | 22.46 | 10.71 | 4.32 | 5.77 | 59.02 | 65.18 | 73.59 | 46.64 |
| **w/ UniPercept Reward** | **22.67** | **10.93** | **4.33** | **6.19** | **65.52** | **74.24** | **77.04** | **59.08** |

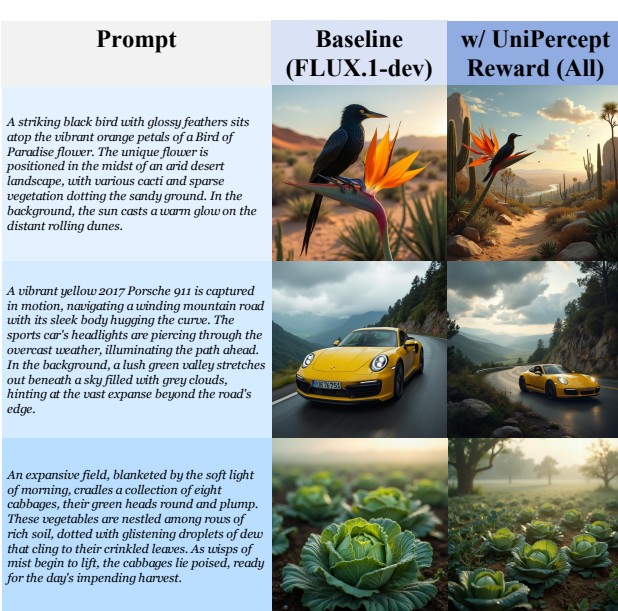

| Prompt | Baseline (FLUX.1-dev) | w/ UniPercept Reward (All) |
|---|---|---|
| *A striking black bird with glossy feathers sits atop the vibrant orange petals of a Bird of Paradise flower. The unique flower is positioned in the midst of an arid desert landscape, with various cacti and sparse vegetation dotting the sandy ground. In the background, the sun casts a warm glow on the distant rolling dunes.* | | |
| *A vibrant yellow 2017 Porsche 911 is captured in motion, navigating a winding mountain road with its sleek body hugging the curve. The sports car's headlights are piercing through the overcast weather, illuminating the path ahead. In the background, a lush green valley stretches out beneath a sky filled with grey clouds, hinting at the vast expanse beyond the road's edge.* | | |
| *An expansive field, blanketed by the soft light of morning, cradles a collection of eight cabbages, their green heads round and plump. These vegetables are nestled among rows of rich soil, dotted with glistening droplets of dew that cling to their crinkled leaves. As wisps of mist begin to lift, the cabbages lie poised, ready for the day's impending harvest.* | | |

*Figure 8.* **Results of FLUX.1-dev w/ UniPercept Reward.**

## 6. Conclusion

We provide **UniPercept-Bench**, a unified benchmark built on a hierarchical definition for perceptual-level image un-

derstanding. We also develop a strong baseline **UniPercept** through Domain-Adaptive Pre-training and Task-Aligned RL, which generalizes well across perceptual domains and outperforms existing MLLMs. UniPercept further serves as a plug-and-play **reward model** for perceptually aligned post-training of T2I models, enabling controllable improvements in perceptual attributes. Beyond model optimization, it also offers a unified perceptual diagnostic tool that reveals systematic behavioral and dataset-level patterns, highlighting its broader utility for future perceptual-level research.

**Limitations.** Although UniPercept-Bench is sufficiently large for current perceptual-level tasks, it remains smaller than typical semantic-level benchmarks. Further expansion in scale will be explored in future work.

## Acknowledgments

This work was supported by Shanghai Artificial Intelligence Laboratory.

## Impact Statement

This paper presents work whose goal is to advance the field of Machine Learning by introducing UniPercept, a unified framework that aligns AI perception with human judgments of aesthetics, quality, and texture. By providing a standardized benchmark and a plug-and-play reward model, our work facilitates the creation of high-fidelity visual content and more intuitive human-AI interaction. We acknowledge that while UniPercept enhances generative capabilities, it also underscores the importance of ethical oversight in managing the risks of hyper-realistic AI-generated imagery and ensuring that aesthetic evaluations remain inclusive of diverse cultural perspectives.

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

# A. Unifying Perceptual-Level Image Understanding

## A.1. From Semantics to Perception

High-level image understanding concerns *what* an image depicts—objects, actions, scenes, and their semantic relations (e.g., "a dog running on grass," "a busy street at night"). In contrast, **perceptual-level image understanding** concerns *how* an image looks and feels to human observers. It captures intrinsic perceptual properties of the visual signal, largely independent of whether semantic content is correctly recognized.

An image may be semantically understandable yet perceptually flawed due to poor composition, compression artifacts, or unrealistic textures. Perceptual-level understanding emphasizes visual cues and their human-perceived effects—beauty and preference, fidelity and distortion, and structural/textural realism—rather than categorical recognition or reasoning. This distinction is well aligned with prior research separating perceptual judgments from semantic interpretation.

We view perceptual-level image understanding as a unified family of human-aligned, low-level perceptual tasks, consisting of:

- **Image Aesthetics Assessment (IAA)**
- **Image Quality Assessment (IQA)**
- **Image Structure & Texture Assessment (ISTA)**

Together, these tasks form a top-down to bottom-up characterization of perceptual attributes:

- **Aesthetics** captures holistic human preference and artistic visual appeal, reflecting composition, mood, and overall expressive quality.
- **Quality** captures visual fidelity and perceived degradation, encompassing distortions and signal-level clarity.
- **Structure & Texture** capture local cues shaping realism and richness, including fine-grained patterns, materials, and small-scale structural details.

This tripartite formulation provides a more complete perceptual description than treating aesthetics, quality, or texture as isolated benchmarks.

## A.2. Detailed Explanation of IAA, IQA, and ISTA

### A.2.1. IMAGE AESTHETICS ASSESSMENT (IAA)

IAA represents the highest level of perceptual abstraction. Its goal is to predict an image's holistic aesthetic appeal—how pleasing, expressive, or artistically valuable it appears.

The task draws on both philosophical aesthetics and empirical studies of human visual preference (Wikipedia contributors, 2025a).

Common aesthetic factors include: (Cao et al., 2025)

- **Composition:** balance, framing, symmetry, rule-of-thirds, leading lines;
- **Visual Elements:** harmony, exposure, atmosphere;
- **Creativity:** intent, novelty, genre coherence;
- **Emotion:** theme communication, storytelling.

Because aesthetic judgments depend strongly on viewer psychology, IAA is the most subjective perceptual dimension.

### A.2.2. IMAGE QUALITY ASSESSMENT (IQA)

IQA bridges subjective perception and objective image formation. Its goal is to estimate perceived visual fidelity, i.e., the severity of degradations relative to an ideal image. Such degradations commonly arise from sensor noise, blur, compression, transmission errors, and artifacts (Wikipedia contributors, 2025b).

Whereas IAA asks "Is this beautiful?", IQA asks "Is this technically clean?" Partially, IQA evaluates: (You et al., 2024; 2025b; Wu et al., 2023a; Chen et al., 2024a)

- **Signal Fidelity:** sharpness, exposure, color naturalness;
- **Artifacts:** blur, noise, compression artifacts, aliasing;
- **Perceptual Faithfulness:** how "undistorted" the image appears.

These properties have both measurable signal correlates and human-dependent thresholds, placing IQA between IAA and ISTA in subjectivity.

### A.2.3. IMAGE STRUCTURE & TEXTURE ASSESSMENT (ISTA)

ISTA measures local perceptual primitives that determine whether an image appears structurally coherent and texturally rich (Wikipedia contributors, 2025c). Unlike the global emphasis of IAA and IQA, ISTA focuses on pixel- and patch-level cues, including:

- **Local Structure:** edges, contours, geometric consistency;
- **Texture Statistics:** granularity, repetitiveness, roughness/smoothness;

- **Material-like Cues:** surface micro-patterns conveying realism (e.g., fabric weave, wood grain).

ISTA connects to classic studies of texture perception and structural similarity, but reframes them as a unified perceptual assessment dimension that many vision models under-represent.

### A.3. The Perceptual Hierarchy

IAA, IQA, and ISTA form a complementary perceptual hierarchy:

- **IAA:** holistic beauty and preference (highly subjective),

- **IQA:** technical fidelity and distortion (mixed objective/subjective),

- **ISTA:** fine-grained structure and texture realism (more objective).

These dimensions are complementary but non-redundant. High quality does not imply high aesthetics; rich texture does not imply high quality; high aesthetics does not imply rich texture.

Thus, no single dimension adequately explains human perceptual judgments.

### A.4. Why Unify Perceptual Understanding?

Unifying IAA, IQA, and ISTA provides key scientific and practical benefits:

- **Shared Perceptual Representations:** joint learning encourages a universal perceptual embedding.

- **Better Evaluators for Generative/Restoration Models:** unified perceptual signals diagnose model failures beyond semantic metrics.

- **Controllable Generation/Editing:** separate perceptual axes support targeted improvements (e.g., enhancing texture without harming quality).

- **Dataset Curation:** perceptual scoring allows filtering or reweighting data by aesthetics, fidelity, or structure.

- **Human-centric Applications:** real-world systems optimize perceived quality, not just semantics.

### A.5. Our Contributions

We introduce a unified paradigm for perceptual-level image understanding. Our contributions include:

- **Unified Task Definition:** a coherent taxonomy integrating IAA, IQA, and ISTA. More details are provided in Tab. 8, Tab. 9, Tab. 10 and Sec. C.3.

- **Holistic Evaluation Protocol:** performance measurement across three perceptual dimensions.

- **Unified Baseline Model:** a multi-task perceptual model predicting aesthetics, quality, and structure/texture jointly.

- **Downstream Validation:** unified perceptual signals improve applications such as reward modeling for post-training.

## B. Details of ISTA

### B.1. Theoretical Basis

**Motivation and Historical Context.** The study (Cimpoi et al., 2014; Scabini et al., 2025) of texture has a long history in computer vision and visual perception. Early works primarily focused on *local texture primitives* and repetitive patterns, emphasizing geometric regularities such as edges, blobs, and their spatial arrangements. Later studies gradually shifted toward *material-oriented representations*, interpreting texture as a visual manifestation of surface properties such as roughness, smoothness, or reflectance. Across these lines of work, texture has traditionally been treated as a *local visual attribute*, defined over a limited spatial region and analyzed independently of global semantics.

**From Local Texture to Instance-aware Attributes.** Building on this tradition, we extend the notion of texture toward *instance-aware structural and textural attributes*, collectively referred to as ISTA. Our key observation is that, in realistic image generation and transformation settings, local visual attributes are rarely independent of the object instance to which they belong. Instead, such attributes are strongly conditioned on object geometry and part-level organization. Concretely, a local region may simultaneously encode: (i) fine-grained surface appearance (e.g., material-related texture), (ii) part-level structural patterns (e.g., circular layouts, layered or repeated arrangements), and (iii) mid-level organization cues that are neither purely geometric nor purely semantic. From this perspective, structure and texture are not arbitrarily merged; rather, they form a *continuum of instance-conditioned local information*, distinct from high-level semantics and global appearance statistics.

**Perceptual and Stylistic Attributes.** Not all instance-related textures correspond directly to physical materials. Certain surface patterns are perceptually defined and human-relevant, such as stylistic regularities or characteristic visual motifs associated with specific object parts. These attributes are consistently recognized by human observers and influ-

ence perceptual judgments, motivating their inclusion in the ISTA category as part of instance-aware local information.

## B.2. Structural Annotation

We establish a systematic definition of image structure and texture, as presented in Tab. 10. Building upon this definition framework, we design Structural Annotation scheme for ISTA as shown in Fig. 9. We also employ the following prompt to guide the MLLM in performing the Structural Annotation for ISTA.

```
{
    "SceneType": "<SceneType>",
    "SceneName": "<SceneName>",
    "Components": [
        {
            "ComponentName": "<Component_1
                >",
            "DescriptionContent": {
                "PhysicalStructure": {
                    "BaseMorphology": ["<
                        Morphology_1>"],
                    "Arrangement": ["<
                        Arrangement_1>"]
                },
                "MaterialRepresentation": {
                    "MaterialClass": ["<
                        MaterialClass_1>"],
                    "SurfaceProperties":
                        ["<
                        SurfaceProperty_1
                        >"]
                },
                "GeometricComposition": {
                    "PlanarContour": ["<
                        PlanarContour_1>"],
                    "VolumetricForm": ["<
                        VolumetricForm_1>"]
                },
                "SemanticPerception": {
                    "FunctionalInference":
                        ["<
                        FunctionalInference_1
                        >"],
                    "StyleType": ["<
                        StyleType_1>"]
                }
            }
        },
        {
            "ComponentName": "<Component_2
                >",
                ...
        }
    }
    ]
}
```

*Figure 9.* Template of ISTA Structural Annotation.

**Prompt for ISTA Structural Annotation**

[PRIOR KNOWLEDGE BASE]
- Base Morphology

blotchy, braided, bubbly, bumpy, chequered, cobwebbed, cracked, crosshatched, crystalline, dotted, fibrous, flecked, freckled, frilly, grid, grooved, honeycombed, interlaced, knitted, lacelike, lined, marbled, matted, meshed, paisley, perforated, pitted, pleated, porous, scaly, smeared, spiralled, sprinkled, stratified, striped, studded, swirly, veined, woven, wrinkled, zigzagged, smooth

- Material Type
1. Natural Materials: Foliage, Grass, Skin, Stone, Wood, Water, Hair
2. Man-Made Materials: Brick, Carpet, Ceramic, Fabric, Glass, Leather, Metal, Mirror, Painted Surface, Paper, Plastic, Polished Stone, Tile, Wallpaper, Concrete, Food Surface
3. Environmental / Background Textures: Sky, Clouds, Fog / Mist

- Two-Dimensional Shape
Rectangle, Square, Circle, Ellipse / Oval, Triangle, Equilateral Triangle, Isosceles Triangle, Scalene Triangle, Right Triangle, Trapezoid / Trapezium, Parallelogram, Rhombus, Pentagon, Hexagon, Heptagon, Octagon, Nonagon, Decagon, Star, Pentagram, Hexagram, Cross, Arrow, Semicircle, Sector, Crescent, Annulus / Ring, Heart, Lemniscate, Lune / Bow Shape, Spiral, Waveform, Teardrop

- Three-Dimensional Shape Categories
Sphere, Ellipsoid, Cube, Cuboid, Cylinder, Cone, Pyramid, Tetrahedron, Octahedron, Dodecahedron, Icosahedron, Prism, Triangular Prism, Rectangular Prism, Pentagonal Prism, Hexagonal Prism, Torus, Annular Torus, Paraboloid, Hyperboloid, Elliptic Cylinder, Hyperbolic Cylinder, Truncated Cone, Truncated Pyramid, Capsule, Dome, Lens, Bipyramid, Frustum, Möbius Strip, Knot, Klein Bottle

- Style Semantics
Embossed, Engraved, Rough, Smooth, Matte, Glossy, Brushed, Honeycomb, Geometric, Fractal, Tile Mosaic, Chinese Cloud Pattern, Dragon Scale, Cyberpunk Holographic, Steampunk Mechanical

[STRUCTURE TEMPLATE]
- Scene Decomposition Principles
A. Single Scene: (Please introduce the description object.)
B. Composite Scene: (Please introduce the different objects. Describe the object separately)
(e.g., street → architectural cluster + pavement system + sky background)

- Description Content (In Composite Scene mode, Please indicate the object in a single description.)
(e.g., Description Content [street])
1. Physical Structure
Base Morphology (*) → Select 1-3 terms from the lexicon (Basically comes from Base Morphology) to describe the surface texture (Focus on texture form rather than shape) Arrangement (!) → Describe spatial layout or directionality of texture: orientation (e.g., diagonal, radial), distribution pattern (e.g., clustered, layered), or density changes (optional)
Dynamics (!) → Motion/transition states (optional)
2. Material Representation
Material Class (*) → Select from Material Type
Surface Properties (!)  → Reflectivity/translucency (optional)
3. Geometric Composition
Planar Contour (!)  → 2D shape terms from Two-Dimensional Shape (where applicable) (optional)
Volumetric Form (!) → 3D form terms from Three-Dimensional Shape Categories (where applicable) (optional)
4. Semantic Perception
Functional Inference (!) → Only!!! with text/icons present (e.g. The icon is: no traffic, and the text is: XX Street...) (optional)
Style Type (!) → Must use style semantics terms (e.g. Baroque style, Xiangyun style...) (optional)
[Execution Standards]
1. Terminology Enforcement: (*)-marked fields require exact term matches
2. Format Purity: Output only structured content, no explanations
3. Hierarchy Preservation: Apply complete template per independent unit
4. Complexity Adaptation: Single description for simple objects, multi-unit decomposition for complex scenes
5. Lexicon Flexibility: For *-marked fields, use official lexicon terms where possible. Free-form extensions are allowed if they add necessary clarity or express phenomena outside the vocabulary.
6. Mixed Mode Expression: Structured descriptions may combine fixed taxonomy terms with precise natural language when facing edge cases or fine-grained observations.

```
[Example]
{
    "SceneType": "Composite Scene",
    "SceneName": "Urban skyscraper
        cluster",
    "Components": [
        {
```

```
            "ComponentName": "Buildings
                ",
            "DescriptionContent": {
                "PhysicalStructure": {
                    "BaseMorphology":
                        ["grid"],
                    "Arrangement": ["
                        Vertical,
                        Symmetrical"],
                    "Dynamics": ["N/A"]
                },
                "MaterialRepresentation
                    ": {
                    "MaterialClass": ["
                        Glass","Metal
                        "],
                    "SurfaceProperties
                        ": ["Glossy","
                        Reflective"]
                },
                "GeometricComposition":
                    {
                    "PlanarContour": ["
                        Rectangle"],
                    "VolumetricForm":
                        ["Cuboid"]
                },
                "SemanticPerception": {
                    "Functional
                    Inference": ["N/A
                        "],
                    "StyleType": ["
                        Modern
                        Architecture"]
                }
            }
        },
        {
            "ComponentName": "Sky
                Background",
            "DescriptionContent": {
                "PhysicalStructure": {
                    "BaseMorphology":
                        ["smooth"],
                    "Arrangement": ["N/
                        A"],
                    "Dynamics": ["N/A"]
                },
                "MaterialRepresentation
                    ": {
                    "MaterialClass": ["
                        Sky"],
                    "SurfaceProperties
                        ": ["N/A"]
                },
                "GeometricComposition":
                    {
                    "PlanarContour": ["
                        N/A"],
                    "VolumetricForm":
                        ["N/A"]
                },
```

```
                "SemanticPerception": {
                    "Functional
                    Inference": ["N/A
                        "],
                    "StyleType": ["N/A
                        "]
                }
            }
        }
    ]
}
```

Please use the above foundational knowledge and the provided example to perform a texture and structural analysis of the image.

### B.3. Definition of ISTA-10K

ISTA-10K is our curated visual rating dataset for evaluating the structure–texture dimension of images. The ISTA score quantifies an image's *structure–texture richness*, reflecting both (1) the **complexity and diversity of its visual textures**, and (2) the **richness and organization of its structural components**. Images exhibiting more varied textures, materials, geometric forms, and semantically distinct elements are assigned higher ISTA scores.

**Texture Intensity Mapping.** To quantify the complexity of base morphological patterns, each texture term $t$ is assigned a discrete weight reflecting its perceived structural richness. Lower weights correspond to visually simple and uniform textures, whereas higher weights indicate more irregular, high-frequency, or structurally complex surface patterns. The weighting function is defined as:

$$w(t) = \begin{cases} 1, & t \in \mathcal{T}_{\text{weak}}, \\ 2, & t \in \mathcal{T}_{\text{medium}}, \\ 3, & t \in \mathcal{T}_{\text{strong}}, \\ 0, & \text{otherwise.} \end{cases} \quad (4)$$

The three texture sets used in Eq. 4 are summarized in Table 6. These categories are derived from commonly observed visual morphologies and are grouped by increasing structural complexity. Weak textures are visually simple and low-variation patterns with minimal geometric or material irregularities. Medium textures exhibit moderate complexity with richer local variations and more structured arrangements. Strong textures reflect high morphological complexity, featuring irregular, multi-scale, or highly distinctive patterns often associated with heterogeneous or fine-grained natural structures.

**Component-Level ISTA Rating.** For each component $c$, we compute:

$$S(c) = S_{\text{PS}}(c) + S_{\text{MR}}(c) + S_{\text{GC}}(c) + S_{\text{SP}}(c). \quad (5)$$

*Table 6.* Texture categories used for texture intensity mapping. Higher groups indicate richer and more complex morphology patterns.

| Category (Weight) | Texture Terms |
|---|---|
| **Weak (1)** | smooth, plain, uniform, lined, grid, striped, chequered, dotted, freckled |
| **Medium (2)** | braided, woven, crosshatched, meshed, cobwebbed, lacelike, knitted, spiralled, swirly |
| **Strong (3)** | bumpy, blotchy, bubbly, cracked, crystalline, flecked, frilly, grooved, honeycombed, marbled, matted, paisley, perforated, pitted, pleated, porous, scaly, smeared, sprinkled, stratified, studded, veined, wrinkled, zigzagged |

These terms quantify different dimensions of structural and perceptual richness:

- **Physical Structure (PS):** describes the surface morphology and spatial arrangement of structural patterns;

- **Material Representation (MR):** captures the diversity of material classes and their surface properties;

- **Geometric Composition (GC):** reflects variations in planar contours and volumetric forms;

- **Semantic Perception (SP):** accounts for functional cues and stylistic categories associated with the component

Each sub-score is defined as follows (all "N/A" entries excluded):

$$S_{\text{PS}}(c) = \sum_{t \in \text{BaseMorphology}(c)} w(t) + \big|\text{Arrangement}(c)\big|, \quad (6)$$

$$S_{\text{MR}}(c) = \big|\text{MaterialClass}(c)\big| + \big|\text{SurfaceProperties}(c)\big|, \quad (7)$$

$$S_{\text{GC}}(c) = \big|\text{PlanarContour}(c)\big| + \big|\text{VolumetricForm}(c)\big|, \quad (8)$$

$$S_{\text{SP}}(c) = \big|\text{FunctionalInference}(c)\big| + \big|\text{StyleType}(c)\big|. \quad (9)$$

**Image-Level ISTA Rating.** If the image contains a set of components $\mathcal{C}$,

$$S_{\text{ISTA}} = |\mathcal{C}| + \sum_{c \in \mathcal{C}} S(c). \quad (10)$$

For images without explicit components, the whole image is treated as a single component:

$$S_{\text{ISTA}} = 1 + S(c_{\text{image}}). \tag{11}$$

Finally, the score is clipped to match the 0–100 rating range:

$$S_{\text{ISTA}} \leftarrow \min(S_{\text{ISTA}}, 100). \tag{12}$$

This formulation yields a deterministic and interpretable measure of structure–texture richness that aligns with the hierarchical annotation schema in UniPercept-Bench.

**Weight of Texture Attributes.** Regarding the weighting function used in Eq. (4), we do not claim a universal psychophysical scale of texture richness. Instead, the weights reflect perceived salience and diagnostic importance of different attributes. Specifically, we rely on human judgments collected from both general volunteers and domain-experienced annotators, who assess which texture attributes are more visually meaningful and more noticeable when degraded. Attributes that are consistently judged as more informative or perceptually significant are assigned higher weights. This weighting scheme serves as a pragmatic approximation of human perceptual priors rather than a theoretically exhaustive model.

### B.4. Human Validation of ISTA Score

To validate whether the proposed ISTA score reflects human perception of structure and texture richness, we conduct a human study on 100 images randomly sampled from UniPercept-Bench. Each image is rated by human volunteers on a 0–10 scale, where a higher score indicates richer structural layouts and more salient texture patterns. Since structure and texture richness can be subjective, we first establish representative examples for high-, medium-, and low-score images to calibrate the rating criteria and reduce annotation ambiguity.

The human ratings show positive correlations with ISTA scores, with $\text{SRCC} = 0.63$ and $\text{PLCC} = 0.67$. These results indicate that ISTA is generally consistent with human perception: images assigned higher ISTA scores tend to be perceived by humans as containing richer structural and textural details. This supports the use of the ISTA score as a quantitative metric for evaluating image structure and texture richness.

## C. Details of UniPercept-Bench

### C.1. Reliability of the Data Construction Pipeline

To address potential model-specific bias in data generation, we conduct a reliability study across several MLLMs, including **GPT-4o** (Achiam et al., 2023), **Gemini-2.5-pro** (Team

et al., 2023), and **Qwen-2.5-VL-72B-Instruct** (Bai et al., 2025). While maintaining identical inputs (source images, professional evaluation, prior definition, and question templates), we first invite a dedicated group of human experts to manually design questions under the same constraints, serving as the gold standard. Subsequently, to assess the quality of both model-generated and expert-crafted data, a separate panel of twenty human volunteers (independent of the question-designing group) performs a blind preference test. They evaluate 10 randomly sampled sets (totaling 200 evaluations) to identify the QA pairs that best align with human perceptual understanding.

*Table 7.* **Human preference rate across different annotators.**

| Annotator | Human Preference Rate |
|---|---|
| Gemini-2.5-pro | 8.0% |
| Qwen-2.5-VL-78B-Instruct | 10.0% |
| GPT-4o | 38.0% |
| Human Expert | 44.0% |
| **Total** | **100.0%** |

As shown in Tab. 7, GPT-4o significantly outperforms other MLLMs and closely approaches the human volunteer baseline. This demonstrates that GPT-4o not only produces high-quality questions but also exhibits strong consistency with human perception. Consequently, we select GPT-4o as the MLLM for initial QA generation.

### C.2. Expert Consensus and Annotation Reliability

As illustrated in Fig. 4, our data construction pipeline relies on **professional evaluation** to provide ground-truth textual descriptions for the generator MLLM. Given the subjectivity of perceptual attributes, we implement a **semantic consensus protocol** to ensure the reliability of these expert annotations.

**Taxonomy-Guided Annotation.** To minimize subjective variance, experts do not write free-form descriptions. Instead, they strictly adhere to the hierarchical *Prior Definition* system (Domain-Category-Criterion) shown in Fig. 4 (a). This constraint forces experts to ground their textual descriptions in standardized perceptual criteria (e.g., using specific terms from the "Material Representation" lexicon rather than ambiguous adjectives), ensuring high consistency across different annotators.

**Double-Blind Verification.** We employ a cross-validation strategy where a primary expert generates the initial textual evaluation, and a secondary expert reviews it against the image. We define "disagreement" as semantic contradictions regarding objective properties (e.g., "matte" versus "glossy") or significant divergence in subjective interpretation (e.g., "chaotic" versus "energetic").

**Consensus and Refinement.** When the secondary expert flags a description as inconsistent, the sample undergoes an offline consultation phase. The experts discuss the divergence to determine if it stems from visual ambiguity or interpretative bias. We only retain the description after both experts reach a consensus on the phrasing. Finally, as shown in Fig. 4 (c), trained volunteers perform a final round of human refinement on the generated QA pairs to ensure they align with the consensus-based professional evaluations.

### C.3. Definition System

The UniPercept definition system is organized into three levels: *Domain–Category–Criterion*. The complete set of fine-grained attributes and their brief descriptions are provided in Tab. 8, Tab. 9 and Tab. 10.

### C.4. Evaluation Details

For the VQA task, we directly feed the image, question, and answer options into the MLLM during evaluation. For the VR task, if a model provides dedicated interfaces for IAA or IQA scoring, we invoke them directly. For models without such interfaces, we evaluate them using the following prompts:

> **Prompt for Visual Rating (IAA)**
>
> Please rate the aesthetics of this image and provide a score between 0 and 100, where 0 represents the lowest quality and 100 represents the highest. Your response should contain only an integer value.

> **Prompt for Visual Rating (IQA)**
>
> Please rate the quality of this image and provide a score between 0 and 100, where 0 represents the lowest quality and 100 represents the highest. Your response should contain only an integer value.

> **Prompt for Visual Rating (ISTA)**
>
> Please rate the structure and texture richness of this image and provide a score between 0 and 100, where 0 represents the lowest quality and 100 represents the highest. Your response should contain only an integer value.

### C.5. Comparison with Other Benchmarks

As shown in Tab. 11, we compare UniPercept-Bench with other widely-used benchmarks in image assessment. UniPercept-Bench provides finer-grained QA categories, supports both rating and textual formats, and employs a

scalable annotation pipeline that combines human expertise with MLLM assistance to achieve example-level annotation. It further covers all three perceptual-level domains and includes both VQA and VR tasks, making it the most comprehensive and well-defined benchmark for perceptual-level image understanding to date.

### C.6. Relations Among Different Perceptual Domains

As discussed in Sec.3.1 of the main paper, the three domains (IAA, IQA, and ISTA) characterize distinct dimensions of image assessment and are therefore largely independent. As illustrated in Fig. 10, an image that receives a high score in IQA may perform poorly in IAA. Likewise, images that achieve strong aesthetic scores may exhibit weaker performance in structural or textural assessment.

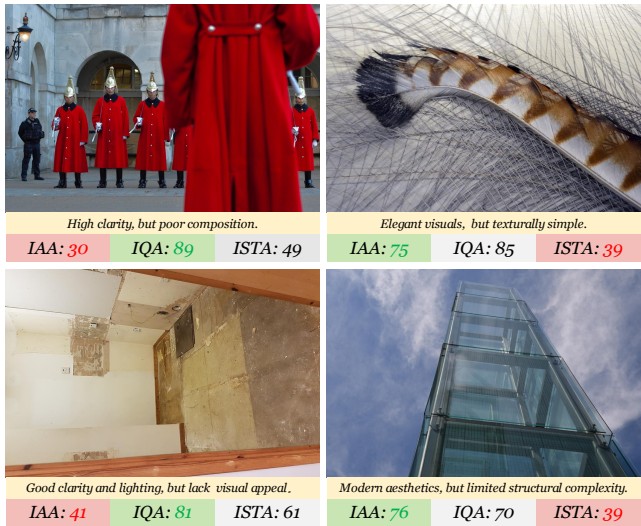

*Figure 10.* **Relationship between different perceptual domains**. Ratings are provided by UniPercept.

## D. Further Discussion on UniPercept

### D.1. Training Dataset

**For Domain-Adaptive Pre-Training.** As shown in Tab. 13, our Domain-Adaptive Pre-Training is conducted on approximately 800K samples spanning the IAA, IQA, and ISTA domains. For the IAA and IQA domains, the data include both text and rating types: the text data consist of high-quality image–text pairs curated from the source datasets using an MLLM (GPT-4o (Achiam et al., 2023)), while the rating data are directly taken from ArtiMuse-10K (Cao et al., 2025) and KonIQ-10K (Hosu et al., 2020). For the ISTA domain, due to the lack of high-quality annotations, we first extract images that meet the domain requirements from the raw datasets and then, following the definition in Sec. B.3, employ GPT-4o to generate structured annotations, which are further converted into textual descriptions. Using all

*Table 8.* **Definition details of IAA domain.**

| No. | Category | Criterion | Description |
|---|---|---|---|
| 1 | Composition & Design (*Comp.*) | Visual Balance | Assess the distribution of visual elements such as shapes, tones, and colors to determine equilibrium across the frame. |
| 2 | Composition & Design | Hierarchical Emphasis | Evaluate the relative prominence of visual elements based on size, contrast, or positioning. |
| 3 | Composition & Design | Structural Organization | Examine spatial alignment, grid conformity, and grouping of elements within the image frame. |
| 4 | Composition & Design | Compositional Rhythm | Assess repetition, spacing, and directional continuity of elements that suggest visual pacing. |
| 5 | Composition & Design | Harmonic Unity | Evaluate visual consistency in shape proportions, relative sizes, and orientation patterns. |
| 6 | Composition & Design | Composition & Design Level | Indicates the overall compositional quality — reflecting balance, rhythm, and structural harmony. |
| 7 | Visual Elements & Structure (*VisStr.*) | Line Dynamics | Assess structure, orientation, and density of linework contributing to visual form. |
| 8 | Visual Elements & Structure | Shape Clarity | Assess clarity of 2D shape boundaries and separation from background. |
| 9 | Visual Elements & Structure | Form Realization | Evaluate rendering of 3D form through shading, lighting gradients, and perspective cues. |
| 10 | Visual Elements & Structure | Spatial Illusion | Judge depth cues such as occlusion, scale variation, and linear perspective. |
| 11 | Visual Elements & Structure | Light Modeling | Assess consistency and realism of lighting, highlights, and shadows. |
| 12 | Visual Elements & Structure | Visual Elements & Structure Level | Measures mastery of fundamental visual elements including lines, shapes, form, and spatial coherence. |
| 13 | Technical Execution (*Tech.*) | Material Proficiency | Evaluate control and precision in using the medium, including brushstroke discipline. |
| 14 | Technical Execution | Rendering Precision | Assess refinement in edge definition, gradient transitions, and micro-level detailing. |
| 15 | Technical Execution | Focus Control | Judge sharpness, blur, or depth-of-field usage to structure visual hierarchy. |
| 16 | Technical Execution | Tonal and Exposure Control | Evaluate luminance distribution and tonal range. |
| 17 | Technical Execution | Technical Execution Level | Represents technical proficiency — tonal control, precision, and rendering quality. |
| 18 | Originality & Creativity (*Creat.*) | Concept Innovation | Assess distinctiveness of concept or narrative. |
| 19 | Originality & Creativity | Creative Problem-Solving | Evaluate ingenuity in visual execution or compositional decision-making. |
| 20 | Originality & Creativity | Originality & Creativity Level | Expresses creative strength — innovation, imagination, and conceptual uniqueness. |
| 21 | Theme & Communication (*Theme.*) | Subject Clarity | Evaluate clarity of subject or message. |
| 22 | Theme & Communication | Narrative Depth | Assess symbolic or narrative layering. |
| 23 | Theme & Communication | Cultural Insight | Judge engagement with cultural, historical, or social ideas. |
| 24 | Theme & Communication | Theme & Communication Level | Evaluates thematic clarity and communication effectiveness. |
| 25 | Emotion & Viewer Response (*Emo.*) | Emotional Resonance | Evaluate emotional tone and viewer response. |
| 26 | Emotion & Viewer Response | Viewer Engagement | Assess long-term viewer compellingness. |
| 27 | Emotion & Viewer Response | Interpretive Openness | Judge clarity–ambiguity balance inviting interpretation. |
| 28 | Emotion & Viewer Response | Emotion & Viewer Response Level | Measures emotional and psychological impact. |
| 29 | Overall Gestalt (*Gest.*) | Holistic Cohesion | Evaluate integration of visual and conceptual components into a unified whole. |
| 30 | Overall Gestalt | Overall Gestalt Level | Represents holistic integration across all components. |
| 31 | Comprehensive Evaluation (*CompEv.*) | Comprehensive Evaluation Level | Synthesizes all artistic dimensions — perceptual quality and conceptual depth. |

*Table 9.* **Definition details of IQA domain.**

| No. | Category | Criterion | Description |
|---|---|---|---|
| 1 | Distortion Location (***Loc.***) | Location Description | Precisely identify and describe specific spatial regions within the image where distortions are visible or concentrated. The question must explicitly reference a concrete part of the image, not the image as a whole. |
| 2 | Distortion Location | Object Association | Specify the semantic or structural objects in the image that are impacted or altered by the distortions. The question must explicitly reference a concrete object, not just a general distortion type. |
| 3 | Distortion Severity (***Sev.***) | Severity Level | Evaluate the severity of distortion in the image as None, Slight, or Obvious, reflecting visibility and perceptual impact. |
| 4 | Distortion Type (***Type.***) | Distortion Types Present | Identify distortion types present in the scene from a comprehensive taxonomy, including blur, noise, compression, brightness, contrast, saturation, sharpening, quantization, exposure, pixelation, color diffusion, jitter, transmission errors, and multi-distortion combinations. |

*Table 10.* **Definition details of ISTA domain.**

| No. | Category | Criterion | Description |
|---|---|---|---|
| 1 | Scene Decomposition Principles (***Scene.***) | Scene Classification | Classify the scene as (A) Single-Object Scene with one primary subject, or (B) Composite Scene containing multiple distinguishable components. Describe the main object(s) clearly. |
| 2 | Physical Structure (***Phys.***) | Base Morphology | Describe surface texture using perceptual descriptors such as fibrous, grooved, marbled, veined, smooth, etc. |
| 3 | Physical Structure | Spatial Arrangement | Describe texture orientation, distribution pattern, and density variation across regions (e.g., horizontal, clustered, layered, radial, uniform). |
| 4 | Material Representation (***Mat.***) | Material Identification | Identify the perceived material category using a standardized taxonomy (natural, man-made, or environmental materials) present in the image. |
| 5 | Material Representation | Surface Behavior | Describe optical surface properties such as glossiness, translucency, or matte finish. |
| 6 | Geometric Composition (***Geo.***) | 2D Contour | Classify the 2D outline shape using a standardized lexicon including basic shapes, polygons, special forms, or organic/curved forms. |
| 7 | Geometric Composition | 3D Volume | Describe the implied 3D volumetric form using a taxonomy of basic solids, polyhedra, prisms, or complex mathematical shapes. |
| 8 | Semantic Perception (***Sem.***) | Functional Suggestion | Infer functional or symbolic implications of textures/motifs based on appearance, referencing standardized functional texture/style descriptors. |
| 9 | Semantic Perception | Stylistic Classification | Assign a stylistic category (e.g., Minimalist, Gothic, Art Deco, Futuristic, Cyberpunk, Chinese Cloud Pattern) based on visual elements and decorative cues. |

*Table 11.* **Comparison with existing benchmarks & datasets.**

| Benchmark | # Test | # QA Category | Data Format | Annotation Level | Annotator | Perceptual-Level Domain | | | Task | |
|---|---|---|---|---|---|---|---|---|---|---|
| | | | | | | *IAA* | *IQA* | *ISTA* | **VQA** | **VR** |
| ***VQA Benchmarks & Datasets*** | | | | | | | | | | |
| Q-Bench (Wu et al., 2023a) | ∼1.5K | – | Text | Category-Level | Human | – | ✓ | – | ✓ | – |
| AesBench (Huang et al., 2024) | ∼10K | 10 | Text | Example-Level | Human | ✓ | – | – | ✓ | – |
| DQ-495K (You et al., 2025b) | ∼56K | – | Text | Category-Level | Human & MLLM | – | ✓ | – | ✓ | – |
| Q-Instruct-DB (Wu et al., 2023b) | – | – | Text | Category-Level | Human & MLLM | – | ✓ | – | ✓ | – |
| Co-Instruct (Wu et al., 2024b) | – | – | Text | Category-Level | Human & MLLM | – | ✓ | – | ✓ | – |
| Q-Ground-100K (Chen et al., 2024a) | ∼1K | – | Text | Category-Level | Human & MLLM | – | ✓ | – | ✓ | – |
| ***VR Benchmarks & Datasets*** | | | | | | | | | | |
| ArtiMuse-10K (Cao et al., 2025) | ∼1K | 15 | Rating & Text | Example-Level | Human | ✓ | – | – | – | ✓ |
| AVA (Murray et al., 2012) | ∼20K | – | Rating | Example-Level | Human | ✓ | – | – | – | ✓ |
| TAD66K (He et al., 2022) | ∼15K | – | Rating | Example-Level | Human | ✓ | – | – | – | ✓ |
| KonIQ-10K (Hosu et al., 2020) | ∼2K | – | Rating | Example-Level | Human | – | ✓ | – | – | ✓ |
| SPAQ (Fang et al., 2020) | ∼1K | – | Rating | Example-Level | Human | – | ✓ | – | – | ✓ |
| KADID (Lin et al., 2019) | ∼1K | – | Rating | Example-Level | Human | – | ✓ | – | – | ✓ |
| **UniPercept-Bench (Ours)** | **∼6K** | **44** | **Rating & Text** | **Example-Level** | **Human & MLLM** | ✓ | ✓ | ✓ | ✓ | ✓ |

*Table 12.* **Ablation studies on UniPercept-Bench-VR.** The best results are highlighted with dark blue cells, and the second-best results with light blue cells. Results are reported as SRCC/PLCC.

| Models | IAA | | IQA | | ISTA |
|---|---|---|---|---|---|
| | ArtiMuse-10K (Cao et al., 2025) | **Avg.** | KonIQ-10K (Hosu et al., 2020) | **Avg.** | ISTA-10K |
| ***Ablation on Training Strategy*** | | | | | |
| w/ Threshold Reward | 0.604/0.556 | 0.617/0.596 | 0.882/0.888 | 0.801/0.790 | 0.303/0.334 |
| w/o Adaptive Pre-Training | 0.546/0.510 | 0.481/0.421 | 0.851/0.817 | 0.733/0.700 | 0.755/0.732 |
| ***Ablation on Training Tasks*** | | | | | |
| VQA-Only | 0.591/0.582 | 0.585/0.598 | 0.816/0.847 | 0.769/0.774 | 0.206/0.206 |
| VR-Only | 0.629/0.596 | 0.558/0.509 | 0.907/0.828 | 0.794/0.749 | 0.767/0.767 |
| ***Ablation on Training Domains*** | | | | | |
| IAA-Only | 0.621/0.608 | 0.508/0.464 | 0.641/0.644 | 0.706/0.680 | 0.197/0.197 |
| IQA-Only | 0.369/0.352 | 0.468/0.435 | 0.901/0.839 | 0.786/0.726 | 0.341/0.337 |
| ISTA-Only | 0.351/0.319 | 0.275/0.288 | 0.595/0.575 | 0.611/0.570 | 0.771/0.782 |
| **UniPercept (Ours)** | **0.746/0.738** | **0.590/0.586** | **0.940/0.949** | **0.824/0.827** | **0.778/0.767** |

*Table 13.* **Data Overview for Domain-Adaptive Pre-Training.**

| Domain | Type | Size | Source |
|---|---|---|---|
| IAA | Text | ~360K | APDDv2 (Jin et al., 2024) Impressions (Kruk et al., 2023) AVA (Murray et al., 2012) TAD66K (He et al., 2022) FLICKR-AES (Ren et al., 2017) |
| | Rating | ~9K | ArtiMuse-10K (Cao et al., 2025) |
| IQA | Text | ~380K | Q-Ground-100K (Chen et al., 2024a) DQ-495K (You et al., 2024) DataDepictQA (You et al., 2024; 2025b) SPAQ (Fang et al., 2020) KADID (Lin et al., 2019) PIPAL (Gu et al., 2020) |
| | Rating | ~7K | KonIQ-10K (Hosu et al., 2020) |
| ISTA | Text | ~40K | DTD (Cimpoi et al., 2014) FMD (Sharan et al., 2014) Big and Small Objects (Konkle & Oliva, 2012) Scene Size x Clutter Database (Park et al., 2015) Reachspaces (Josephs et al., 2021) Flickr2K (Lim et al., 2017), LSDIR (Li et al., 2023) |
| | Structure | ~40K | *Same as Text* |

the above data, we perform Domain-Adaptive Pre-Training to equip UniPercept with an initial capacity for perceptual-level image understanding.

*Table 14.* **Data Overview for Task-Aligned RL.**

| Domain | Type | Size | Source |
|---|---|---|---|
| IAA | VR | ~9K | ArtiMuse-10K (Cao et al., 2025) |
| | VQA | ~10K | UniPercept Data-VQA (train) |
| IQA | VR | ~7K | KonIQ-10K (Hosu et al., 2020) |
| | VQA | ~10K | UniPercept Data-VQA (train) |
| ISTA | VR | ~10K | ISTA-10K |
| | VQA | ~10K | UniPercept Data-VQA (train) |

**For Task-Aligned RL.** As shown in Tab. 14, we summarize the data composition used in the Task-Aligned RL stage. For the three domains (IAA, IQA, and ISTA), we combine data from both the VR and VQA tasks. For the VR task, we use the training sets from ArtiMuse-10K, KonIQ-10K, and ISTA-10K. For the VQA task, we construct training data following the same pipeline described in Fig. 4 of the main paper, and denote the resulting training set as

*UniPercept Data-VQA (train).* By jointly leveraging all six types of training sources, the Task-Aligned RL stage enables UniPercept to effectively handle both **two** task formats and all **three** perceptual domains.

## D.2. Ablation Studies

We further investigate the training of UniPercept and evaluate its performance on both the VR and VQA dimensions of UniPercept-Bench. The results are reported in Tab.12 and Tab.15.

*Table 15.* **Ablation studies on UniPercept-Bench-VQA.**

| Experiments | IAA | IQA | ISTA | Avg. |
|---|---|---|---|---|
| ***Ablation on Training Strategy*** | | | | |
| w/ Threshold-based Reward | 72.32% | 76.29% | 81.65% | 76.75% |
| w/o Adaptive Pre-Training | 69.16% | 75.09% | 80.00% | 74.75% |
| ***Ablation on Training Tasks*** | | | | |
| VQA-Only | 71.92% | 76.29% | 81.44% | 76.55% |
| VR-Only | 68.57% | 68.38% | 75.15% | 70.70% |
| ***Ablation on Training Domains*** | | | | |
| IAA-Only | 73.69% | 69.67% | 75.57% | 72.98% |
| IQA-Only | 64.73% | 76.01% | 77.53% | 72.76% |
| ISTA-Only | 69.56% | 69.58% | 82.27% | 73.80% |
| **UniPercept (Ours)** | **76.55%** | **81.07%** | **84.23%** | **80.62%** |

### D.2.1. TRAINING STRATEGY

**Domain-Adaptive Pre-Training.** We empirically verify the importance of Domain-Adaptive Pre-Training. As shown in Tab.12 and Tab.15, removing Domain-Adaptive Pre-Training leads to a substantial performance drop on both the VR and VQA tasks. This indicates that a vanilla MLLM initialization exhibits limited perceptual-level understanding, and that training on sufficiently large, domain-relevant data is necessary to equip the model with fundamental perceptual capabilities.

**Reward Design.** For the VR task, we also consider a threshold-based reward, following the formulation used in Q-Insight (Li et al., 2025b), which determines correctness solely based on the numerical deviation between the predic-

*Table 16.* **More Performance Results of FLUX.1-dev w/ UniPercept Reward.**

| Models | Preference Score | | Image Quality | Image Aesthetics | | UniPercept Score | | |
|---|---|---|---|---|---|---|---|---|
| | PickScore (Kirstain et al., 2023) | HPSv3 (Ma et al., 2025) | DeQA (You et al., 2025a) | LAION-Aes (Schuhmann & Beaumont, 2022) | ArtiMuse (Cao et al., 2025) | IAA | IQA | ISTA |
| Baseline | 22.46 | 10.71 | 4.32 | 5.77 | 59.02 | 65.18 | 73.59 | 46.64 |
| w/ UniPercept IAA Reward | 22.47 | 10.09 | 4.09 | 6.19 | 67.02 | 76.20 | 76.39 | 54.83 |
| w/ UniPercept IQA Reward | 22.63 | 11.21 | 4.37 | 6.02 | 63.64 | 72.16 | 76.87 | 52.34 |
| w/ UniPercept ISTA Reward | 22.72 | 11.09 | 4.37 | 6.16 | 63.75 | 72.23 | 76.17 | 59.61 |
| **w/ UniPercept Reward (All)** | **22.67** | **10.93** | **4.33** | **6.19** | **65.52** | **74.24** | **77.04** | **59.08** |

tion and the ground truth. Let $p_i$ and $g_i$ denote the predicted and ground-truth scores. A prediction is considered correct if its absolute error falls within a predefined tolerance $\epsilon$, and incorrect otherwise. This produces a binary reward signal that avoids extreme reward magnitudes and encourages predictions to stay within an acceptable range:

$$r_{\text{thr}}^{(i)} = \begin{cases} 1, & \text{if } |p_i - g_i| < \epsilon, \\ 0, & \text{otherwise.} \end{cases} \quad (13)$$

In our experiments, we set the threshold to match the one used by the Adaptive Gaussian Soft Reward in UniPercept. As shown in Tab.12 and Tab.15, the Adaptive Gaussian Soft Reward consistently outperforms the threshold-based formulation and even leads to improvements on the VQA task, where rating rewards are not directly applied. These results highlight the advantages of the Adaptive Gaussian Soft Reward in more accurately capturing the deviation between predicted and ground-truth scores. They also indicate that the VQA and VR tasks share underlying correlations, such that advances in one task can benefit the other.

### D.2.2. TRAINING DATA

**Multi-Task vs. Single-Task.** We further investigate the relationship between the VR and VQA tasks. During both Domain-Adaptive Pre-Training and Task-Aligned RL, we separate the VR and VQA training data while keeping all other text data unchanged, and conduct two settings: VQA-only and VR-only. As shown in Tab.12 and Tab.15, the VQA-only model performs poorly on the VR task, and the VR-only model similarly performs poorly on the VQA task. Moreover, both settings underperform UniPercept (VR & VQA) even on their respective target tasks. These results demonstrate that jointly training on both VR and VQA tasks provides substantial mutual benefits and leads to stronger perceptual understanding.

**Multi-Domain vs Single-Domain.** We also study the relationships among the three perceptual domains: IAA, IQA, and ISTA. During both Domain-Adaptive Pre-Training and Task-Aligned RL, we separate the training data of the three domains while keeping all other data unchanged, and conduct three single-domain settings: IAA-only, IQA-only, and ISTA-only. As shown in Tab.12 and Tab.15, models trained on a single domain perform well on their corresponding domain but fall short in overall performance. In contrast,

UniPercept (trained with a mixture of all three domains) achieves substantially stronger overall results and even surpasses the single-domain models on certain tasks. These findings indicate that, although IAA, IQA, and ISTA focus on different aspects of perceptual assessment, jointly training on all three domains enhances the model's holistic perceptual understanding.

### D.3. Semantic vs. Perceptual Capability

As suggested by prior work, we believe that downstream training may lead to some change in pretrained capabilities. To examine semantic capability after perceptual training, we additionally evaluate UniPercept and its base model InternVL3-8B (Zhu et al., 2025) on MMBench (Liu et al., 2024a), MMMU (Yue et al., 2024a), and MMStar (Chen et al., 2024c). UniPercept shows a **mild drop** in general capability, but the magnitude is limited.

*Table 17.* **Semantic performance after perceptual training.**

| | MMBench (en) | MMMU (val) | MMStar |
|---|---|---|---|
| InternVL3-8B | 86.00 | 58.67 | 68.40 |
| UniPercept | 83.33 | 52.00 | 64.87 |

UniPercept-Bench still requires substantial semantic understanding, as its QA pairs are semantically coherent and several perceptual categories rely on higher-level scene understanding beyond local appearance cues. Overall, this suggests that **perceptual** and **semantic** abilities are not independent, and perceptual-domain training does **not** cause severe semantic degradation, though a mild trade-off may exist.

### D.4. UniPercept Reward

We further investigate the use of UniPercept as a reward model for text-to-image generation. Specifically, we employ the IAA Rating, IQA Rating, and ISTA Rating from UniPercept as three separate reward signals and conduct post-training under the exact same settings as Tab.5 in the main paper. More performance comparison is reported in Tab. 16.

We observe that each reward emphasizes **a different aspect** of image assessment, leading to distinct performance advantages: training with the *IAA reward* yields improve-

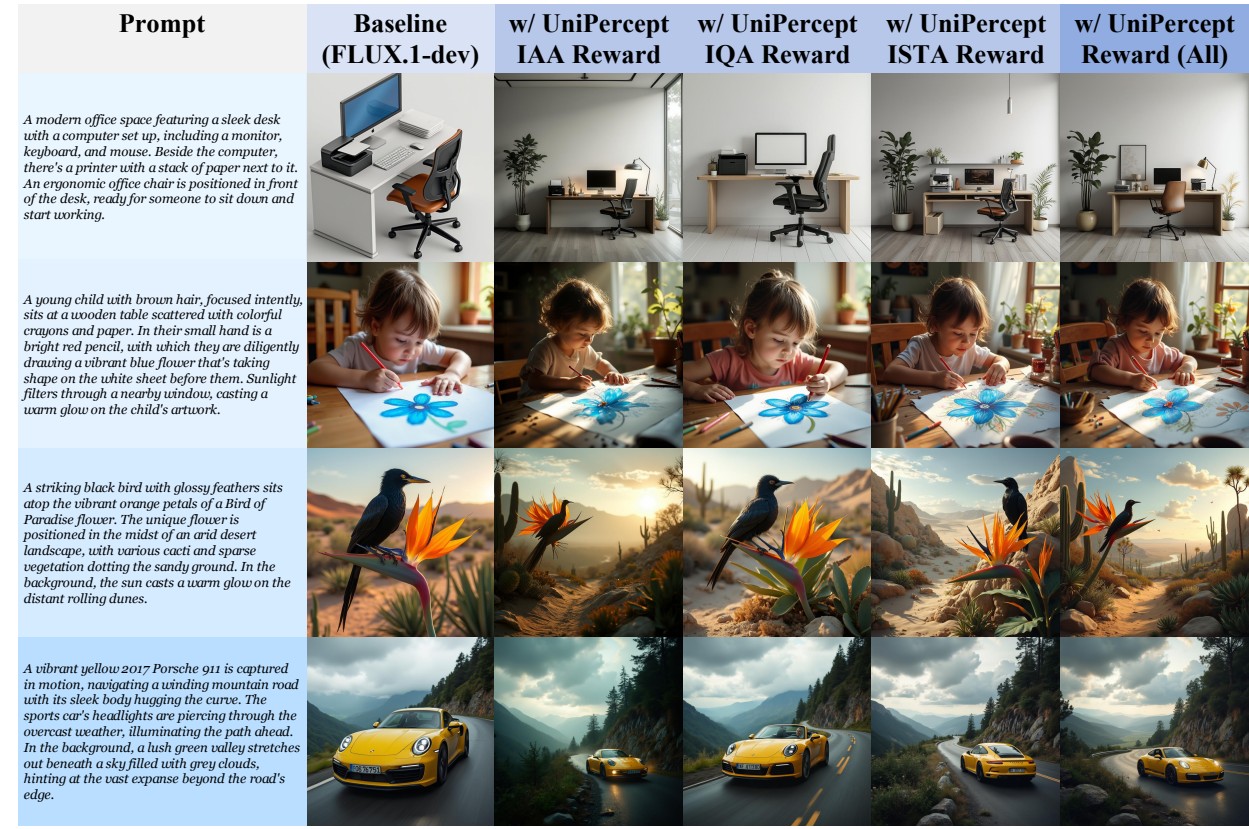

*Figure 11.* **More results of FLUX.1-dev w/ UniPercept Reward.** Different reward signals emphasize distinct perceptual attributes, while *UniPercept Reward (All)* achieves the best overall performance by integrating complementary perceptual cues.

*Table 18.* **UniPercept Metrics on Various Datasets.**

| Dataset | UniPercept-IAA | UniPercept-IQA | UniPercept-ISTA | Avg. |
|---|---|---|---|---|
| ***Natural Images*** | | | | |
| ImageNet (Russakovsky et al., 2015) | 53.88 | 61.90 | 36.79 | 50.85 |
| Unsplash (Unsplash, 2025) | 62.49 | 69.19 | 43.32 | 58.33 |
| DF2K (Timofte et al., 2017; 2018a;b; Ignatov et al., 2019) | 45.99 | 52.92 | 34.78 | 44.56 |
| LAION-5B (Schuhmann et al., 2022) | 60.56 | 69.21 | 38.85 | 56.21 |
| ***AIGC Images*** | | | | |
| Blip3o-60K (Chen et al., 2025) | 63.81 | 73.88 | 49.38 | 62.36 |
| ImgEdit (Ye et al., 2025) | 55.83 | 59.77 | 36.88 | 50.83 |

ments primarily on aesthetics-related metrics, while the *IQA reward* leads to stronger performance on quality-oriented metrics. In contrast, the *ISTA reward* provides more balanced gains across all dimensions. Using the full *UniPercept reward (All)* integrates the strengths of all three domains and achieves the best overall performance. Additional qualitative comparisons are provided in Fig. 11.

### D.5. UniPercept Metrics

**For Models.** UniPercept can serve as an perceptual-level metric that assesses the quality of outputs from any model producing images, covering three complementary dimensions: IAA, IQA, and ISTA. Here, we take text-to-image models as an example and apply UniPercept as an evaluation metric on both the DPG (Hu et al., 2024) and

GenEval (Ghosh et al., 2023) benchmarks to measure the perceptual quality of generated images. Except for special cases (e.g., GPT-Image-1 (Achiam et al., 2023), where output resolution cannot be controlled), all models are evaluated with a fixed output resolution of 1024×1024. The results are reported in Tab.19 and Tab.20.

We observe that while current models generally achieve strong performance in IQA, there remains **substantial room for improvement in IAA and ISTA**. Moreover, as model capability advances, we see consistent improvements in both instruction-following ability (as reflected by DPG and GenEval metrics) and perceptual image quality (as reflected by UniPercept). For instance, GPT-Image-1 (Achiam et al., 2023) and Qwen-Image (Wu et al., 2025a) exhibit strong performance across both Dataset Metrics and UniPercept Metrics. More detailed analyses will be presented in future

*Table 19.* **Evaluation of T2I Models on DPG (Hu et al., 2024) Metrics and UniPercept Metrics.**

| Models | DPG Metrics (Hu et al., 2024) | | | | | | UniPercept Metrics | | | |
|---|---|---|---|---|---|---|---|---|---|---|
| | Global | Entity | Attribute | Relation | Other | Overall | IAA | IQA | ISTA | Avg. |
| OmniGen (Xiao et al., 2024) | – | – | – | – | – | – | 62.83 | 72.22 | 45.09 | 60.04 |
| OmniGen2 (Wu et al., 2025b) | 88.81 | 88.83 | 90.18 | 89.37 | 90.27 | 83.57 | 58.51 | 71.89 | 43.31 | 57.90 |
| BAGEL (Deng et al., 2025) | 88.94 | 90.37 | 91.29 | 90.82 | 88.67 | 85.07 | 60.20 | 70.52 | 45.78 | 58.83 |
| SANA-1.6B (Xie et al., 2024; 2025) | 86.00 | 91.50 | 88.90 | 91.90 | 90.70 | 84.80 | 40.33 | 42.89 | 42.41 | 41.87 |
| Lumina-DiMOO (Xin et al., 2025) | 81.46 | 92.08 | 88.98 | 94.31 | 82.00 | 86.04 | 61.00 | 71.14 | 44.83 | 58.99 |
| FLUX.1-dev (Labs et al., 2025) | 74.35 | 90.00 | 88.96 | 90.87 | 88.33 | 83.84 | 65.18 | 73.59 | 46.64 | 61.80 |
| GPT-Image-1 (Achiam et al., 2023) | 88.89 | 88.94 | 89.84 | 92.63 | 90.96 | 85.15 | 62.27 | 72.87 | 44.88 | 60.00 |
| Qwen-Image (Wu et al., 2025a) | 91.32 | 91.56 | 92.02 | 94.31 | 92.73 | 88.32 | 62.89 | 72.15 | 47.23 | 60.76 |

*Table 20.* **Evaluation of T2I Models on GenEval (Ghosh et al., 2023) Metrics and UniPercept Metrics.**

| Models | GenEval (Ghosh et al., 2023) Metrics | | | | | | | UniPercept Metrics | | | |
|---|---|---|---|---|---|---|---|---|---|---|---|
| | Single Obj. | Two Obj. | Counting | Colors | Position | Attr. Bind. | Overall | IAA | IQA | ISTA | Avg. |
| OmniGen (Xiao et al., 2024) | 0.99 | 0.86 | 0.64 | 0.85 | 0.31 | 0.55 | 0.70 | 58.84 | 75.62 | 41.00 | 58.49 |
| OmniGen2 (Wu et al., 2025b) | 0.99 | 0.96 | 0.74 | 0.98 | 0.71 | 0.75 | 0.86 | 54.20 | 75.16 | 34.48 | 54.61 |
| BAGEL (Deng et al., 2025) | 0.99 | 0.94 | 0.81 | 0.88 | 0.64 | 0.63 | 0.82 | 58.68 | 71.24 | 38.35 | 56.09 |
| SANA-1.6B (Xie et al., 2024; 2025) | 0.99 | 0.77 | 0.62 | 0.88 | 0.21 | 0.47 | 0.66 | 34.34 | 35.11 | 31.22 | 33.56 |
| Lumina-DiMOO (Xin et al., 2025) | 1.00 | 0.94 | 0.85 | 0.89 | 0.85 | 0.76 | 0.88 | 51.93 | 71.98 | 30.86 | 51.59 |
| FLUX.1-dev (Labs et al., 2025) | 0.98 | 0.81 | 0.74 | 0.79 | 0.22 | 0.45 | 0.66 | 64.24 | 74.96 | 41.14 | 60.11 |
| GPT-Image-1 (Achiam et al., 2023) | 0.99 | 0.92 | 0.85 | 0.92 | 0.75 | 0.61 | 0.84 | 69.07 | 76.74 | 51.26 | 65.69 |
| Qwen-Image (Wu et al., 2025a) | 0.99 | 0.92 | 0.89 | 0.88 | 0.76 | 0.77 | 0.87 | 52.02 | 74.44 | 34.13 | 53.53 |

work.

**For Datasets.** We evaluate a variety of natural-image and AIGC-image datasets using UniPercept as the assessment metric, with results summarized in Tab.18. Among all datasets, Unsplash (Unsplash, 2025) and Blip3o-60K(Chen et al., 2025) achieve the strongest overall performance across the three domains of IAA, IQA, and ISTA. A more in-depth investigation of these distributional characteristics is left for future work.

# E. More Examples of UniPercept-Bench

We provide additional examples from UniPercept-Bench in Fig. 12.

# F. UniPercept-Constructed Image Profiles

UniPercept is capable of performing comprehensive perceptual-level analysis of images, providing accurate visual-rating evaluations across the IAA, IQA, and ISTA dimensions, together with fine-grained, multi-dimensional analytical outputs. This enables UniPercept to generate a detailed *profile* for each image. We present examples of **UniPercept-Constructed Image Profiles** in Fig. 13, Fig. 14, and Fig. 15.

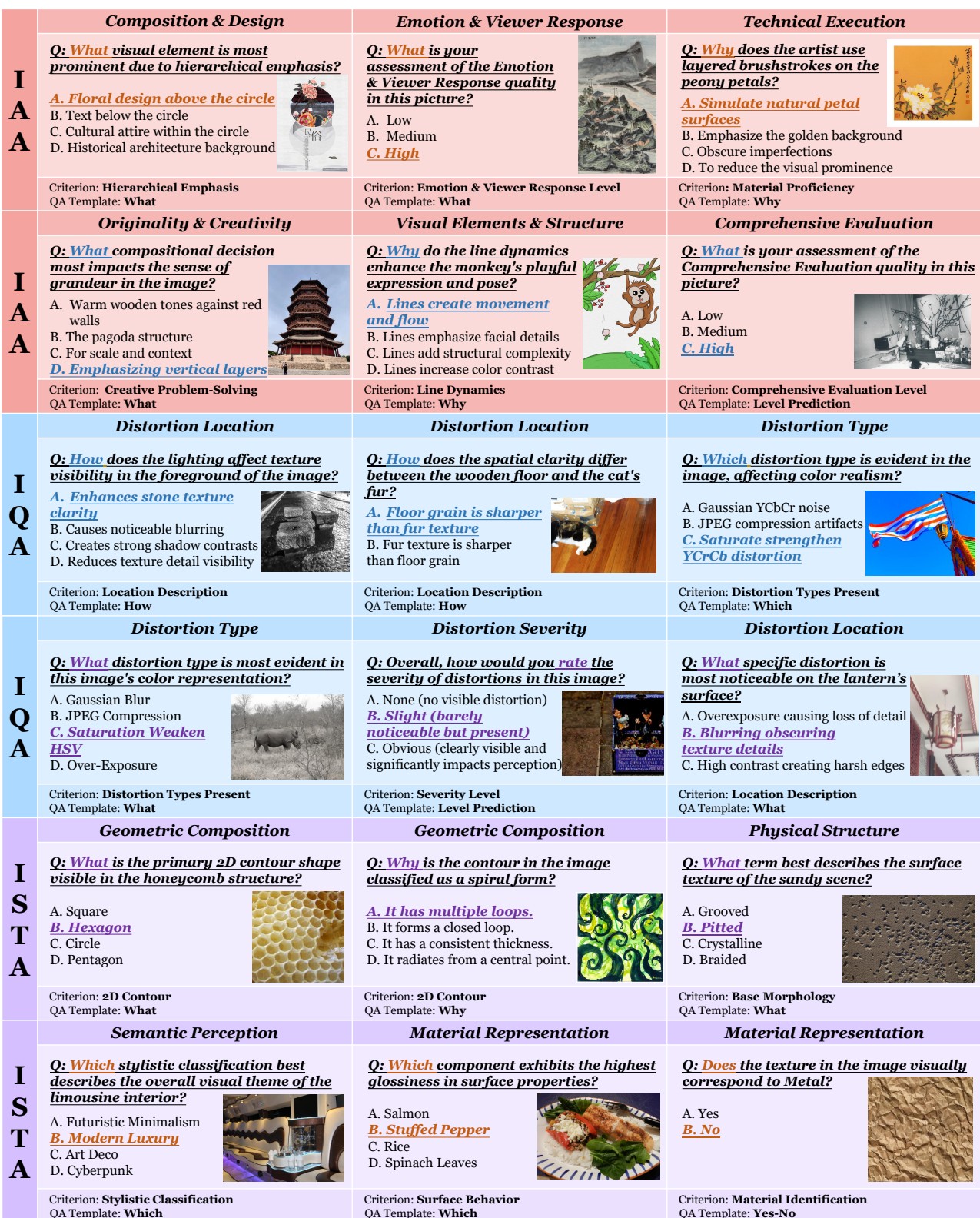

*Figure 12.* **More Examples in UniPercept-Bench.**

# 📝 *UniPercept-Constructed Image Profile*

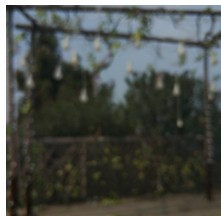

**IAA Rating: 29/100**

**IQA Rating: 35/100**

**ISTA Rating: 46/100**

## Image Aesthetics Assessment (IAA)

**Compostion & Design**
- *The composition is nonexistent, with a completely blurred and indistinct subject ...*
- *There is no balance or rhythm to speak of. The elements are randomly scattered ...*

**Technical Execution**
- *The image exhibits catastrophic technical execution.*
- *Every possible mistake is present, with extreme blurriness rendering all details invisible.*
- *The focus is nonexistent, resulting in a complete lack of sharpnessness.*

...

**Comprehensive Evaluation**
- *The image suffers from extreme blurriness, rendering any aesthetic evaluation impossible.*
- *The lack of focus eliminates any potential visual impact, technical merit, or depth. Every element, from the pergola to the lanterns, is indistinct, failing to convey any meaningful ...*
- *In summary, the image's total absence of visual coherence results in a zero aesthetic merit.*

## Image Quality Assessment (IQA)

**Distortion Location**
- *The entire scene, including the pergola, lanterns, and background foliage, is out of focus, resulting in a complete loss of sharpnessness. This blurring affects every element uniformly, making it impossible to identify ...*
- *The image fails to convey any meaningful content due to its total lack of clarity.*

**Distortion Severity**
- *The image suffers from extreme blurriness.*
- *This catastrophic blurring negates any possible evaluation of other distortion types, as nothing is visible.*
- *The final distortion severity is the maximum imaginable.*

...

**Overall Quality**
- *The image is severely out of focus, rendering all elements indistinct and lacking any discernible detail.*
- *The lack of sharpnessness, exposure, and any visible content results in a complete failure in image quality. This extreme blurriness negates any possible merit, making it impossible to evaluate any positive attributes.*
- *In summary, the image quality is catastrophic, with zero merit due to total blurriness.*

## Image Structure & Texture Assessment (ISTA)

**Scene Decomposition Principles**
- **Scene Type**: *Blurred Composite Scene*
- **Scene Name**: *Garden with pergola and hanging lights*

**Componet: Pergola**
- **Physical Structure**: *[Base Morphology: Lined] [Arrangement: Horizontal]*
- **Material Representation**: *[Material Class: Glass] [Surface Properties: Glossy]*
- **Geometric Composition**: *[Volumetric Form: Sphere]*

**Componet: Background Foliage**
- **Physical Structure**: *[Base Morphology: Matted] [Arrangement: Layered]*
- **Material Representation**: *[Material Class: Foliage] [Surface Properties: Matte]*
- **Geometric Composition**: *[Volumetric Form: Naturalistic]*

**Componet: Hanging Lights**
- **Physical Structure**: *[Base Morphology: Smooth] [Arrangement: Clustered]*
- **Material Representation**: *[Material Class: Glass] [Surface Properties: Glossy]*
- **Geometric Composition**: *[Volumetric Form: Sphere]*

...

*Figure 13.* **Example of UniPercept-Constructed Image Profiles.**

# 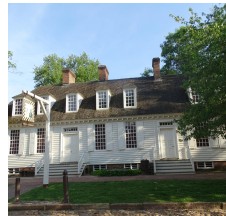 *UniPercept-Constructed Image Profile*

| IAA Rating: 87/100 |
| --- |

| IQA Rating: 75/100 |
| --- |

| ISTA Rating: 56/100 |
| --- |

## Image Aesthetics Assessment (IAA)

**Originality & Creativity**
- *The image depicts a well-preserved historical house with a straightforward and classic presentation.*
- *While the composition and clarity are excellent, the photograph adheres closely to traditional documentation styles, lacking experimental or imaginative elements.*

**Emotion & Viewer Response**
- *The image evokes a strong sense of nostalgia and admiration for historical architecture, effectively capturing ...*
- *The well-preserved colonial house and lush green create a tranquil, inviting atmosphere, likely sparking curiosity and appreciation for historical preservation. While the composition is perfect and visually pleasing, it may ...*

*...*

**Comprehensive Evaluation**
- *The image showcases a well-preserved historical house with strong visual impact due to its symmetrical ...*
- *Overall, the image excels in historical representation but slightly lacks dynamic engagement. The aesthetic merit is strong, reflecting both historical and visual excellence.*

## Image Quality Assessment (IQA)

**Distortion Type / Location / Severity**
- *There is no distortion in the image.*

*...*

**Overall Quality**
- *The image showcases a well-preserved historical building with excellent clarity and vibrant colors, capturing the architectural details and surrounding green effectively.*
- *The composition is balanced, with well-lit, sharp focus, and excellent exposure.*
- *The cobblestone path and lush trees enhance the scene's authenticity.*
- *There are no visible flaws, indicating perfect sharpnessness and color accuracy.*
- *This image excels in every aspect, achieving perfection.*
- *Overall, this is a flawless, high-quality photograph.*

## Image Structure & Texture Assessment (ISTA)

**Scene Decomposition Principles**
- **Scene Type**: *Composite Scene*
- **Scene Name**: *Historic house with surrounding environment*

**Componet: House**
- **Physical Structure**: *[Base Morphology: Lined, Smooth] [Arrangement: Horizontal]*
- **Material Representation**: *[Material Class: Wood] [Surface Properties: Matte]*
- **Geometric Composition**: *[Planar Contour: Rectangle] [Volumetric Form: Cuboid]*
- **Semantic Perception**: *[Style Type: Colonial Architecture]*

**Componet: Roof**
- **Physical Structure**: *[Base Morphology: Tiled] [Arrangement: Layered]*
- **Material Representation**: *[Material Class: Tile] [Surface Properties: Matte]*
- **Geometric Composition**: *[Planar Contour: Rectangle]*
- **Semantic Perception**: *[Style Type: Traditional]*

**Componet: Chimneys**
- **Physical Structure**: *[Base Morphology: Brick] [Arrangement: Vertical]*
- **Material Representation**: *[Material Class: Brick] [Surface Properties: Matte]*
- **Geometric Composition**: *[Planar Contour: Rectangle] [Volumetric Form: Cuboid]*
- **Semantic Perception**: *[Style Type: Traditional]*

*...*

*Figure 14.* **Example of UniPercept-Constructed Image Profiles.**

# 📝 *UniPercept-Constructed Image Profile*

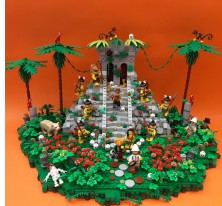

| IAA Rating: 75/100 |
|---|

| IQA Rating: 81/100 |
|---|

| ISTA Rating: 82/100 |
|---|

## Image Aesthetics Assessment (IAA)

**Visual Elements & Structure**
- *The image excels in visual elements and structure, showcasing a harmonious blend of vibrant colors ...*
- *The lush green of the jungle contrasts beautifully with the bright orange background, enhancing visual impact. The composition is meticulously balanced, with dynamic focal points like the pyramid and characters, creating a cohesive and engaging scene.*

**Theme & Communication**
- *The image masterfully communicates the theme of a jungle tomb raid through its vivid and intricate LEGO ...*
- *Every element, from the pyramid to the characters and animals, is meticulously crafted to evoke a sense of adventure and danger. The narrative is clear and engaging, effectively capturing the essence of Indiana Jones.*

**Overall Gestalt**
- *The image showcases a masterpiece of LEGO artistry, seamlessly blending every element to perfection.*
- *The overall gestalt is nothing short of perfection, with every detail meticulously crafted to create a cohesive, enchanting jungle scene. The vibrant colors, intricate details, and harmonious composition evoke a sense ...*

*...*

## Image Quality Assessment (IQA)

**Distortion Type / Location / Severity**
- *There is no distortion in the image.*

*...*

**Overall Quality**
- *The image is sharp and well-composed, showcasing a detailed LEGO jungle scene without any distortion.*
- *The vibrant colors and intricate details are intact, with no signs of blurring, overexposure, or compression artifacts. The scene's perfection is evident in its clarity and balance. There are no visual flaws.*
- *The lighting and composition are flawless, enhancing the intricate details and depth.*
- *There are no visible flaws, making it a flawless representation.*
- *This image excels beyond perfection, meriting a perfect score. In summary, the image is flawless in every aspect, showcasing impeccable quality.*

## Image Structure & Texture Assessment (ISTA)

**Scene Decomposition Principles**
- **Scene Type***: Composite Scene*
- **Scene Name***: Lego Jungle Temple*

**Componet: Temple Structure**
- **Physical Structure***: [Base Morphology: Blocky, Grid] [Arrangement: Layered]*
- **Material Representation***: [Material Class: Plastic] [Surface Properties: Matte]*
- **Geometric Composition***: [Planar Contour: Rectangle] [Volumetric Form: Cuboid]*

**Componet: Palm Trees**
- **Physical Structure***: [Base Morphology: Fibrous, Frilly] [Arrangement: Vertical]*
- **Material Representation***: [Material Class: Plastic] [Surface Properties: Matte]*
- **Geometric Composition***: [Volumetric Form: Cylinder]*

**Componet: Flora**
- **Physical Structure***: [Base Morphology: Matted, Frilly] [Arrangement: Clustered]*
- **Material Representation***: [Material Class: Plastic] [Surface Properties: Matte]*

**Componet: Figures**
- **Physical Structure***: [Base Morphology: Blocky] [Arrangement: Clustered]*
- **Material Representation***: [Material Class: Plastic] [Surface Properties: Matte]*

*...*

*Figure 15.* **Example of UniPercept-Constructed Image Profiles.**

