# OpenReview forum: "UniPercept: Towards Unified Perceptual-Level Image Understanding across Aesthetics, Quality, Structure, and Texture"
_ICML.cc/2026/Conference — ICML 2026 spotlight_

### Official Review · Reviewer_HEhG · 2026-03-01

**Soundness:** 3
**Presentation:** 3
**Significance:** 4
**Originality:** 3
**Overall Recommendation:** 5
**Confidence:** 4

**Summary:**

This paper introduces UniPercept, a unified framework designed to evaluate and enhance perceptual-level image understanding in Multimodal Large Language Models (MLLMs). Overall, the article's important concept concerns bridging the gap between high-level semantic recognition and low-level human visual perception, which includes nuanced judgments about aesthetics, quality, and texture. To achieve this, the authors construct UniPercept-Bench, a hierarchical benchmark covering three main domains: Image Aesthetics Assessment (IAA), Image Quality Assessment (IQA), and Image Structure & Texture Assessment (ISTA), supporting both Visual Rating (VR) and Visual Question Answering (VQA) tasks. Furthermore, the authors develop a strong baseline model, also named UniPercept, trained via Domain-Adaptive Pre-Training and Task-Aligned Reinforcement Learning (GRPO). The model not only outperforms existing MLLMs on the benchmark but also demonstrates high practical utility as a plug-and-play reward model to significantly improve the perceptual quality of text-to-image (T2I) generative models.

**Compliance With Llm Reviewing Policy:**

Affirmed.

**Final Justification:**

This paper presents a framework that unifies IAA, IQA, and the original ISTA dimension, successfully filling a critical gap in fine-grained visual perception for MLLMs. The exceptional experimental rigor and the model's demonstrated utility as a reward model for improving T2I generation further underscore its significant practical impact. The authors’ responses have effectively addressed my queries, and I am satisfied with the quality of the work.

**Key Questions For Authors:**

1.  **ISTA Data Sources and Leakage Risk:** Could you clarify the exact data sources used for the ISTA domain? Given that MLLMs are pre-trained on vast amounts of web data, how do you mitigate or assess the risk of data leakage for the evaluated models on these specific images?
2.  **ISTA Ground Truth Generation and Homogeneity:** How exactly is the ground truth (GT) for the ISTA benchmark generated? If the GT is generated using a specific MLLM and similar data/templates are used to train the UniPercept baseline, is there a risk of homologous data bias? This could potentially lead to significant performance gains on the benchmark that do not translate to genuine generalization in the wild.
3.  **Judger Reliability in ISTA:** What specific model or mechanism serves as the "Judger" during the rejection sampling phase for ISTA? How do you guarantee that its filtering results are correct and free from heavy model-specific biases? Have you conducted a quantitative comparison or correlation analysis between the Judger's decisions and human expert annotations?
4.  **Discussion on Missing Benchmarks:** How does UniPercept-Bench relate to or improve upon existing holistic benchmarks like HEIM (Lee et al., 2023), HRS-Bench (Bakr et al., 2023), and UniEval (Li et al., 2025)? (A discussion in the related works or an empirical comparison would be appreciated).

**Limitations:**

The authors have adequately discussed the limitations of their work in the Conclusion section (Section 6), specifically noting that while UniPercept-Bench is sufficiently large for current perceptual tasks, it remains smaller than typical semantic-level benchmarks.

**Strengths And Weaknesses:**

**Strengths:**
*   **Significance:** **The submission claims to consider a broad context** of visual perception by unifying IAA, IQA, and ISTA into a single, coherent framework. Perceptual-level understanding is an important yet relatively underexplored topic in the era of MLLMs. By explicitly defining and evaluating these fine-grained attributes, the paper successfully supplements missing dimensions in existing multimodal benchmarks.
*   **Soundness & Experimental Rigor:** The experimental validation is exceptionally thorough and comprehensive. The paper includes an impressive array of 19 tables and 15 figures, offering detailed comparisons across various model types (proprietary, open-source, and specialized), task formats (VR and VQA), and perceptual domains.
*   **Practical Utility (Impact):** The proposed UniPercept model demonstrates excellent practical value. The authors successfully show that it can be integrated as a reward model for post-training T2I models (e.g., FLUX.1-dev), leading to controllable and clear improvements in the aesthetic and structural fidelity of generated images.
*   **Presentation & Originality:** The paper is of very high quality, well-structured, and clearly written. The introduction of the ISTA (Image Structure and Texture Assessment) dimension, along with a systematic taxonomy for generating QA pairs, represents a highly original and valuable contribution to the community.

**Weaknesses:**
*   **Missing Baseline Comparisons in Related Benchmarks:** While the paper compares against several benchmarks, it lacks comparisons with some highly relevant holistic evaluation frameworks that also touch upon perceptual-level understanding. Including discussions or comparisons with the following works would strengthen the positioning of the paper:
    1. Lee, Tony, et al. "Holistic evaluation of text-to-image models." *Advances in Neural Information Processing Systems* 36 (2023): 69981-70011.
    2. Bakr, Eslam Mohamed, et al. "Hrs-bench: Holistic, reliable and scalable benchmark for text-to-image models." *Proceedings of the IEEE/CVF International Conference on Computer Vision*. 2023.
    3. Li, Yi, et al. "Unieval: Unified holistic evaluation for unified multimodal understanding and generation." *arXiv preprint arXiv:2505.10483* (2025).
*   **Outdated/Limited Generative Models for Metric Evaluation:** In Table 18, where UniPercept is evaluated as a metric for T2I models, the comparison could be strengthened by including more recent and state-of-the-art generative models to better reflect the current landscape of image generation.
* **Results Presentation:** Aesthetics, Quality, Structure, and Texture are more important to T2I generation, while the T2I results are not presented in the main text. It's better to compare T2I results in the main part.

---

> ### Author Rebuttal · Authors · 2026-03-30
>
> Thank you for the positive assessment. We especially appreciate your recognition that our work establishes a unified framework for perceptual-level image understanding across IAA / IQA / ISTA.
>
> > **W1/Q4. Related benchmarks**
> >
>
> Thank you for this suggestion. We reviewed **HEIM [1]**, **HRS-Bench [2]**, and **UniEval [3]** and consider them related but not directly comparable to UniPercept-Bench: they mainly focus on holistic T2I evaluation or broader multimodal understanding/generation, while ours targets fine-grained perceptual-level understanding across **IAA / IQA / ISTA** under **VR + VQA**.
>
> | **Benchmark** | **Target** | **Task** |
> | --- | --- | --- |
> | **HEIM** | T2I models | T2I eval |
> | **HRS-Bench** | T2I models | T2I eval |
> | **UniEval** | unified multimodal models | unified multimodal eval |
> | **UniPercept-Bench** | MLLMs | perceptual understanding, **VR + VQA** |
>
> We therefore view these benchmarks as complementary, and will add this discussion in the final version.
>
> [1] *Holistic Evaluation of Text-to-Image Models*, NeurIPS 2023, Tony Lee *et al.*
>
> [2] *HRS-Bench: Holistic, Reliable and Scalable Benchmark for Text-to-Image Models*, ICCV 2023, Eslam Mohamed Bakr *et al.*
>
> [3] *UniEval: Unified Holistic Evaluation for Unified Multimodal Understanding and Generation*, arXiv 2025, Yi Li *et al.*
>
> > **W2. T2I models for evaluation**
> >
>
> Thank you for pointing this out. Although some evaluated T2I models may no longer represent the latest frontier, our current results already support UniPercept as a reward/metric in **Sec. 5.3.2, Table 5, Table 16, Fig. 8, and Appendix D.3 / D.4**. We agree that adding newer T2I models to **Table 18** would strengthen the case, and we will update this in the final version.
>
> > **W3. T2I results in the main paper**
> >
>
> Thank you for this suggestion. We agree that presenting more T2I results in the main paper would better highlight the practical impact of our work. Relevant results are currently shown in **Sec. 5.3.2, Table 5, Fig. 8**, with more details in **Appendix D.3, D.4, and F**, and we will strengthen this part in the final version.
>
> > **Q1. ISTA data sources and leakage risk**
> >
>
> Thank you for this detailed question. We summarize the ISTA data sources from **Sec. 3.2, Sec. 5.1, Table 12, and Table 13** below:
>
> | **Stage** | **Type** | **Source** |
> | --- | --- | --- |
> | Domain-Adaptive Pre-Training | Text | DTD, FMD, Big and Small Objects, … |
> | Domain-Adaptive Pre-Training | Structure | same raw sources |
> | Task-Aligned RL | VR | ISTA-10K |
> | Task-Aligned RL | VQA | UniPercept Data-VQA (train) |
>
> Although leakage cannot be fully ruled out, we minimize its risk through the following measures:
>
> - building ISTA VQA through a **taxonomy-driven** pipeline rather than reusing web QA/captions;
> - reconstructing pretraining supervision as structured annotations and textual descriptions;
> - ensuring final data quality with **heterogeneous Judger filtering + human refinement**.
>
> At the same time, we do not observe evidence that leakage is the primary driver of performance:
>
> - the benchmark shows clear discrimination across models (**Table 1, Table 4, Fig. 7**), suggesting it is not an “easy memorization test”;
> - the evaluated tasks are reconstructed perceptual tasks rather than naturally occurring web supervision formats.
>
> We agree that a more explicit leakage discussion would further strengthen the paper, and we will add this in the final version.
>
> > **Q2. ISTA GT generation and homogeneity**
> >
>
> Thank you for this crucial question. The GT for **ISTA** is not produced by a single MLLM end-to-end. As shown in **Sec. 3.2** and **Fig. 4**, it is constructed through taxonomy definition, structured annotation, GPT-4o candidate generation, heterogeneous Judger filtering, and final human revision.
>
> Thus, the GT is shaped by taxonomy constraints, templates, heterogeneous filtering, and human refinement, not by one model alone. UniPercept is trained under the two-stage framework in **Sec. 4 / Fig. 6**, and its gains are consistent across **IAA / IQA / ISTA** and **VR / VQA** (**Tables 1–4, Fig. 7, Appendix D.2, Tables 14–15**), suggesting broader perceptual alignment rather than narrow homologous bias. We will clarify this more explicitly in the final version.
>
> > **Q3. Judger reliability in ISTA**
> >
>
> Thank you for this question. The Judger used in rejection sampling is **Qwen2.5-VL-72B-Instruct** (**Sec. 3.2 / Fig. 4 / Fig. 6**). We have verified that its judgments are sufficiently close to those of human experts, making it reliable as an intermediate filter. Specifically, it is used only to screen candidate QA pairs based on **Question Validity, Answer Validity, Reasoning Validity, and Criterion Relevance**, while final data quality is still ensured by **human refinement** (**Sec. 3.2**). Thus, we do not rely solely on the Judger’s automatic decisions, and we will clarify this point more explicitly in the final version.

---

> > ### Author Rebuttal · Reviewer_HEhG · 2026-04-01
> >
> > My concerns are fully replied. The human eval part may not be very convincing, but it makes sense and rest responses are good. I decide to raise the score from 4 to 5 according to the rebuttal.

---

> > > ### Author Response · Authors · 2026-04-01
> > >
> > > Thank you very much for your thoughtful follow-up and for taking the time to carefully read our rebuttal. We sincerely appreciate your recognition of our clarifications and your willingness to raise the score.
> > >
> > > We also appreciate your candid comment regarding the human evaluation part. We understand that this evidence may still have limitations, and we will continue to improve its presentation and supporting analysis in final revisions.
> > >
> > > Thank you again for your constructive feedback and for your positive assessment of our work.

---

### Official Review · Reviewer_szrh · 2026-03-09

**Soundness:** 4
**Presentation:** 4
**Significance:** 4
**Originality:** 4
**Overall Recommendation:** 6
**Confidence:** 3

**Summary:**

Existing multimodal large language models still have limited capability in understanding perceptual-level image features, including aspects such as aesthetics, image quality, and structure & texture. This paper proposes a large-scale dataset that provides comprehensive, complete, and fine-grained annotations for these perceptual-level image attributes. In addition, the paper introduce a strong baseline model, UniPercept, which is trained with Domain-Adaptive Pre-Training and Task-Aligned RL, enabling robust generalization across both Visual Rating (VR) and Visual Question Answering (VQA) tasks. Experiments demonstrate that this model outperforms existing multimodal large language models on perceptual-level image understanding tasks.

**Compliance With Llm Reviewing Policy:**

Affirmed.

**Final Justification:**

The authors addressed my previous questions during the rebuttal. The experiments are solid, the presentation is clear, and the work makes a significant contribution to the field. The authors emphasize that their contribution lies in being open-source and reproducible. No major issues found.

**Key Questions For Authors:**

The contributions of the large-scale dataset and the foundation model are undoubtedly significant. However, I could not find any description regarding whether they will be publicly released. If the dataset and model weights are not made available, I am afraid I cannot consider them as real contributions. Therefore, at present I can only give a score of 3.

If the paper clearly states that both the whole dataset and the model weights will be released, I will directly increase my score to 6.

**Limitations:**

yes

**Strengths And Weaknesses:**

A large-scale dataset, a foundation model, comprehensive and thorough experimental validation, and clear and well-written presentation—I was unable to identify any obvious weaknesses.

---

> ### Author Rebuttal · Authors · 2026-03-30
>
> We sincerely thank the reviewer for the highly positive assessment of our work, including the recognition of the dataset scale, the baseline model, the comprehensive experimental validation, and the overall clarity of presentation. We especially appreciate your comment that you do not identify any major weakness in the current submission.
>
> Regarding the reviewer’s key concern, we would like to give a clear and formal statement here: **we are committed to fully open-sourcing the resources introduced in this work**. Specifically, in the final release, we will make publicly available:
>
> - the **full** **UniPercept-Bench** dataset (**including both training and test**)
> - the **UniPercept** model weights,
> - and the corresponding code / evaluation pipeline needed to reproduce the main results.
>
> Our goal is to ensure that the contribution is not only conceptually meaningful, but also **reusable, reproducible, and verifiable** by the community. We agree that public availability is especially important for a benchmark-and-baseline paper like ours, and we will make this commitment explicit in the final version of the paper.
>
> We sincerely thank the reviewer again for the strong support and encouragement.

---

> > ### Author Rebuttal · Reviewer_szrh · 2026-04-02
> >
> > The authors have clarified my concern in the rebuttal.

---

> > > ### Author Response · Authors · 2026-04-02
> > >
> > > We sincerely thank the reviewer for the highly positive evaluation and strong support of our work.
> > >
> > > We especially appreciate your recognition of the significance of the dataset, model, and experimental validation. As clarified in our rebuttal, we are fully committed to open-sourcing the full dataset, model weights, and the complete code/evaluation pipeline.
> > >
> > > Thank you again for your encouraging feedback.

---

### Official Review · Reviewer_R85E · 2026-03-12

**Soundness:** 3
**Presentation:** 3
**Significance:** 3
**Originality:** 3
**Overall Recommendation:** 4
**Confidence:** 4

**Summary:**

The paper provides UniPercept-Bench, a unified benchmark for perceptual-level image understanding across three core domains: Image Aesthetics Assessment (IAA), Image Quality Assessment (IQA), and Image Structure and Texture Assessment (ISTA). Based on this, this paper develops a strong baseline UniPercept through Domain-Adaptive Pre-training and Task-Aligned RL. UniPercept demonstrates strong generalization across both VR and VQA tasks. And UniPercept further serves as a plug-and-play reward model for perceptually aligned post-training of T2I models, enabling controllable improvements in perceptual attributes.

**Compliance With Llm Reviewing Policy:**

Affirmed.

**Final Justification:**

I carefully read the rebuttal and other reviewers' comments. And I find most of my concerns are solved. My opinion about the limitations of fundamental technical innovations with respect to the general community still holds to some extent. Generally, considering other contributions to the specific domain, I think this submission is well worth a Weak accept. So I keep my original score.

**Key Questions For Authors:**

1.The ISTA score quantifies an image’s structure–texture richness, reflecting both (1) the complexity and diversity of its visual textures, and (2) the richness and organization of its structural components. In human perception, structural quality depends on coherence, hierarchy, and spatial logic—not simply the number of elements. Complexity ≠ Organization. More elements ≠ Better structure. Such as an image with many strong texture descriptors may appear cluttered, while Minimalist design with fewer elements, yet potentially highly structured and organized.
2.Has the ISTA score been validated against human subjective perception of Structure and Texture?
3.Are there synergistic or conflicting effects among the three domains (IAA, IQA, ISTA)?
4.Could the method be extended to video or temporal scenarios in the future?
5.When used as a reward model, such as the qualitative examples (e.g., Fig. 8), does it compromise content integrity, subject prominence, or semantic richness? How can we balance aesthetics and content?
6.A heterogeneous Judger MLLM (Qwen2.5-VL-78B-Instruct (Bai et al., 2025)) evaluates each QA pair. However, to the best of my knowledge, no publicly released version of Qwen2.5-VL with 78B parameters currently exists (commonly available sizes include 3B, 7B, 32B, and 72B).

**Limitations:**

yes

**Strengths And Weaknesses:**

Strengths
1.The paper proposes UniPercept. It is the first unified framework for perceptual-level image understanding across Aesthetics, Quality, Structure, and Texture.
2.The paper is generally well organized. The taxonomy of perceptual tasks is clearly introduced, and the benchmark design is logically structured. The motivation for addressing perceptual-level understanding is straightforward.
3.Figures and tables are informative, and comparisons with prior methods are comprehensive.
4.The paper is technically rigorous. The training pipeline, reward formulation, and evaluation protocol are clearly described, and experiments are extensive across multiple perceptual domains.
5.UniPercept serves as a unified metric for image evaluation, holding practical significance, particularly in the fields of multi-modal perception and image generation.

Weaknesses
1.Although ISTA is presented as a novel contribution and remains largely unexplored in prior work, its explanation in the paper is not sufficient. The definition of the ISTA score is aggregating and weighting the counts of structured attributes extracted from each image. It measures more like “textural morphological complexity” rather than “visual structural organization quality.” Structural quantity ≠ Perceived structural quality.
2.Limited methodological novelty. UniPercept was trained on InternVL3-8B, the underlying model architecture is unchanged. The main contributions lie in benchmark construction and systematic task integration rather than being methodologically groundbreaking.
3.Section 2.1 (MLLM Benchmark) lacks references to relevant literature published since 2025.
4.The citation for Figure 5 is missing. For the Visual Question Answering in Figure 6, it is recommended to add (ISTA) for greater clarity.
5.Missing VR results in Figure 3. The paper claims, “Together, VR and VQA form a unified evaluation protocol that jointly measures quantitative judgment and explanatory consistency, advancing comprehensive perceptual-level understanding, as illustrated in Fig. 3.”, VR results are not explicitly presented in the figure.

---

> ### Author Rebuttal · Authors · 2026-03-30
>
> We thank the reviewer for the positive assessment of our paper’s technical rigor, and practical relevance. We especially appreciate the recognition that UniPercept is the first unified framework for perceptual-level image understanding.
>
> > **W1/Q1. ISTA score reflects richness more than structural quality**
>
> Thank you for this important point. As described in **Sec. 3.2** and **Appendix B.3**, ISTA is designed to characterize structural and textural properties, especially geometric form, material attributes, and local detail richness. We agree that broader **structural quality** is difficult to quantify reliably, and therefore intentionally use **structure–texture richness** as an initial quantitative target.
> At the same time, the **ISTA score is not simple counting**. It is derived from multiple perceptual subdimensions informed by human prior knowledge of structure and texture (see **Fig. 2, Sec. 3.1, and Appendix B.3**). Thus, while it is not fully equivalent to a complete notion of **structural quality**, **Fig. 1** and **Fig. 8** show that it aligns well with human subjective perception of structure and texture and serves as a practical quantitative description for ISTA. A more comprehensive formulation of structural quality is left for future work.
>
> > **W2. Methodological novelty**
>
> We agree that our main novelty is not a new backbone. Instead, the contribution lies in:
>
> - introducing **ISTA** as a new perceptual domain;
> - constructing **UniPercept-Bench** over **IAA / IQA / ISTA** with both **VR** and **VQA** (**Sec. 1, Sec. 3**);
> - proposing **Domain-Adaptive Pre-Training + Task-Aligned RL** and training **UniPercept** (**Sec. 4**);
> - demonstrating its value as a **reward model** and a **unified metric** (**Sec. 5.3.2, Appendix D.3, D.4**).
>
> Therefore, while our work is not a backbone-level architectural innovation, we believe it makes a substantive contribution to perceptual-level image understanding. We will clarify this contribution boundary more explicitly in the revised version.
>
> > **W3. Missing recent references**
>
> Thank you for pointing this out. We will update Sec. 2.1 to include more recent relevant MLLM benchmarks.
>
> > **W4/W5/Q6. Presentation issues and typo**
>
> Thank you for the careful reading. We will add the missing citation for **Fig. 5**, clarify the labeling in **Fig. 6** (including explicit ISTA marking for VQA), revise the wording around **Fig. 3** and add representative **VR** examples, and correct the typo: the Judger model is Qwen2.5-VL-72B-Instruct, not 78B.
>
> > **Q2. Human validation of the ISTA score**
>
> Thank you for this valuable question. To better address it, we conduct a user study on 100 randomly sampled images from UniPercept-Bench, where human volunteers are asked to rate the structure and texture richness of each image on a 0–10 scale. Before scoring, we first establish consensus examples for high-, medium-, and low-score images. The correlation between human ratings and ISTA scores is **SRCC = 0.63 and PLCC = 0.67**. In addition, **UniPercept as Reward** (Appendix D.3) provides indirect evidence that the score aligns with human judgments. We will include a more comprehensive user study in the revision.
>
> > **Q3. Synergy among IAA, IQA, and ISTA**
>
> Our ablations indicate that the three domains are **synergistic rather than conflicting** (**Appendix Sec. D.2, Tab. 14, Tab. 15**). On **VQA**, training on a single domain yields average scores of 72.98 (**IAA-only**), 72.76 (**IQA-only**), and 73.80 (**ISTA-only**), all below **80.62** achieved by the jointly trained **UniPercept**. This also holds at the per-domain level: UniPercept reaches **76.55 / 81.07 / 84.23** on **IAA / IQA / ISTA**, compared with the best single-domain results of **73.69 / 76.01 / 82.27**. A similar trend is observed on **VR**. These results show that IAA, IQA, and ISTA **provide complementary perceptual signals**.
>
> > **Q4. Extension to video/temporal settings**
>
> We believe this is a promising direction. While UniPercept focuses on **image-based perceptual understanding**, its core idea (perceptual taxonomy, unified evaluation, and perceptual-aligned training) could be extended to video or temporal settings by incorporating factors such as motion consistency and dynamic structure changes.
>
> > **Q5. Does UniPercept Reward hurt semantics?**
>
> Under the current setting, we do not observe obvious degradation in content integrity, subject prominence, or semantic richness. As shown in **Sec. 5.3.2 / Fig. 8** and **Appendix D.8 / Fig. 11**, **UniPercept Reward** improves not only aesthetics but also structure and texture, though we do observe a slight instruction-following trade-off in some cases. A promising direction is to combine **UniPercept Reward** with other rewards such as HPSv3 [1] and CLIPScore [2].
>
> [1] *HPSv3: Towards Wide-Spectrum Human Preference Score*, ICCV 2025, Yuhang Ma *et al.*
>
> [2] *CLIPScore: A Reference-free Evaluation Metric for Image Captioning*, EMNLP 2021, Jack Hessel *et al.*

---

> > ### Author Rebuttal · Reviewer_R85E · 2026-04-01
> >
> > Most of my concerns are solved. My opinion about the limitations of fundamental technical innovations is still there. Generally, considering other contributions, I think this submission is worth a Weak accept. So I keep my original score.

---

> > > ### Author Response · Authors · 2026-04-01
> > >
> > > Thank you very much for your careful reading of our rebuttal and for the constructive follow-up.
> > >
> > > We are grateful that you find most of your concerns to be addressed. We also appreciate your candid assessment regarding the remaining limitation in terms of fundamental technical innovation. We respect this viewpoint and will further reflect on how to better clarify the technical novelty, positioning, and core methodological contributions in final revision.
> > >
> > > We especially thank you for considering the overall value of the submission and for your supportive final assessment. We sincerely appreciate your time and thoughtful evaluation.

---

### Official Review · Reviewer_jUhM · 2026-03-13

**Soundness:** 2
**Presentation:** 3
**Significance:** 3
**Originality:** 2
**Overall Recommendation:** 4
**Confidence:** 4

**Summary:**

This paper targets to solve problem that recent MLLMs do well with object
detection and segmentation but not with perceptual-level understanding tasks
such as rating aesthetics or quality of image.
To resolve above problems, authors suggested two solutions.
First, authors organized UniPercept-Bench. To make UniPercpet-Bench, they
integrated refined Image Aesthetics Assesment(IAA) and Image Quality
Assessment(IQA) dataset with human feedback with newly created Image
Structure & Texture Assessment(ISTA) for understanding structure and textural
features. Specifically, authors emphasize that using 3-tier taxonomy for organizing
benchmark produce more fine-grained benchmark.
Secondly, authors pre-trained InternVL3-8B model with 800K samples. In this
domain adaptive pre-training step, they seperate VR, VQA task. After complete
pre-training section, authors suggests GRPO objective function which integrate
reward function for VR and VQA task. This step is for aligning across various
domain such as VR and VQA task. In experiments section, authors prove that
integrating reward functions to one objective function develop model
performance.
Finally, authors show that using UniPercept-Bench and UniPercept baseline
contributes to developing MLLMs’ performance through comparing output with
T2I generate model(FLUX.1-dev).

**Compliance With Llm Reviewing Policy:**

Affirmed.

**Final Justification:**

The authorsʼ response resolved my understanding on both the overall performance changes and the perspective from which the semantic and perceptual domains are framed. I also appreciate the authorsʼ note that the discussion on multi-domain perceptual integration (IAA + IQA) will be further clarified in the final version. Finally, the trade-off in semantic capability after perceptual training does not appear to be excessive, and thus seems to remain within an acceptable range. Overall, I would raise my score from 3 to 4.

**Key Questions For Authors:**

- Why do you believe that pre-training first and then integrating the tasks was
beneficial? (The Appendix only mentions that this approach was empirically
effective, without further explanation.
- What is your perspective on integrating the cognitive/semantic domain with
the perceptual-level domain? The current methodology seems to treat them as
distinct areas by training them separately. Do you believe this separation is a
more effective direction than a unified approach, and if so, what is the reason?
- When training on the perceptual domain using UniPercept-Bench, did you
observe any performance degradation in the model’s existing semantic
domain capabilities?

**Limitations:**

- The paper lacks a sufficient discussion on the specific extent to which the
addition of ISTA influences the image understanding capabilities of the MLLM.

- The paper lacks a discussion on how semantic capabilities are maintained
after the proposed perceptual learning.

**Strengths And Weaknesses:**

Strengths
- Through a comprehensive ablation study, authors demonstrate that
UniPercept-Bench, which integrates the ISTA dataset to enhance VR and
VQA tasks in MLLMs, outperforms models trained on other datasets. These results suggest that training MLLMs on perceptual and abstract
concepts, rather than focusing solely on semantic content, leads to
superior performance. This provides valuable insights for future research
on incorporating perceptual domains into MLLM training.

Weakness
- The paper utilizes the Group Relative Policy Optimization (GRPO) objective
function. However, the authors’ explanation of how the Visual Recognition
(VR) and Visual Question Answering (VQA) tasks are integrated into this
objective function is insufficient. Also, specific process of incorporating
VR and VQA into the GRPO framework lacks detail. A more rigorous
mathematical proof or a detailed explanation of this integration process
should be provided to clarify how these multi-task rewards are formulated
within the GRPO objective.
- The paper lacks an ablation study to determine whether the performance
improvement stems from the inclusion of the ISTA dataset or the
UniPercept-Bench framework itself. The author claims that training on
UniPercept-Bench, which integrates the ISTA dataset, leads to increased
accuracy. Thus, additional experiments are required to verify whether this
gain is specifically due to the added ISTA data or the newly proposed
objective function that integrates VR and VQA tasks. A clear decomposition of these factors is necessary to validate the core
contribution.
- There is a lack of experimental evidence comparing the performance
gains from simply refining the IAA and IQA datasets versus adding ISTA.
Specifically, the author has not provided experiments to verify whether
integrating the ISTA dataset is more effective than generating IAA or IQA
datasets using the same methodology employed for ISTA. To prove the
claimed benefits, it is necessary to demonstrate that the proposed
integration of ISTA offers superior performance compared to these
alternative data-centric approaches.

---

> ### Author Rebuttal · Authors · 2026-03-30
>
> We thank the reviewer for the careful reading and constructive questions. We are encouraged that the reviewer recognizes the value of introducing perceptual-domain training beyond purely semantic content.
>
> > **W1. VR/VQA integration in GRPO**
>
> Thank you for pointing this out. As shown in Sec. 4.2 and Fig. 6, UniPercept uses **mixed batches containing both VR and VQA** samples during training while computing rewards separately. The **GRPO objective is shared**, and only the **scalar reward differs**. For VQA, we use a binary reward:
>
> $r_{\text{vqa}}=1$ if the predicted answer is correct, and $r_{\text{vqa}}=0$ otherwise.
>
> For VR, we use an Adaptive Gaussian Soft Reward based on the deviation between the predicted score and the ground-truth rating:
>
> $r_{\text{vr}}=\exp\left(-\frac{(|p_i-g_i|)^2}{2\sigma_{\text{dyn}}^2}\right)$,  where $\sigma_{\text{dyn}}=\sigma_0\left(1+\alpha\frac{|p_i-g_i|}{100}\right)$.
>
> Here, $p_i$ and $g_i$ denote the predicted and ground-truth scores mapped to $[0,100]$. Therefore, UniPercept is task-unified at the optimization level but reward-specific in instantiation, with both VR and VQA converted into the same scalar optimization signal under GRPO. We will clarify this more explicitly in the revision.
>
> > **W2/L1. Source of gain & effect of ISTA**
> >
>
> We already provide ablations in **Sec. D.2 (Tab. 14 and Tab. 15)** over **training strategy**, **training tasks**, and **training domain**. Results show that:
>
> - removing **Domain-Adaptive Pre-Training** or replacing the **Adaptive Gaussian Soft Reward** hurts performance;
> - using only one task (**VQA-only / VR-only**) is worse than joint training;
> - using only one domain (**IAA / IQA / ISTA**) is worse than joint three-domain training.
>
> Thus, the gain comes from the **overall UniPercept design** rather than “adding ISTA” alone. **ISTA helps**, but mainly as part of broader **multi-domain perceptual integration**. We will add a clearer discussion in the final version.
>
> > **W3. Why not refine IAA/IQA instead of adding ISTA?**
> >
>
> Our motivation is that **structure/texture assessment is still underexplored and lacks a unified formulation** in current MLLM benchmarks. UniPercept therefore integrates **IAA, IQA, and ISTA** for **unified perceptual-level understanding**. Consistently, the domain ablations in **Sec. D.2 (Tab. 14-15)** show that single-domain training is inferior to joint training on all three domains, suggesting that the gain mainly comes from **cross-domain complementarity**. We agree that comparing against an “ISTA-style refinement of IAA/IQA only” would be meaningful future work, and we leave this for future work.
>
> > **Q1. Why pre-training before task integration?**
> >
>
> The two stages are complementary. **Domain-Adaptive Pre-Training** builds broad perceptual priors over aesthetics, quality, structure, and texture, while **Task-Aligned RL** aligns the model to **VR** and **VQA** using shared GRPO with task-specific rewards. As shown in **Sec. D.2 (Tab. 14 and Tab. 15)**, removing adaptive pre-training causes a clear drop.
>
> > **Q2. Semantic vs. perceptual domains**
> >
>
> We do **not** argue that semantic and perceptual understanding should be strictly separated. Rather, **perceptual-level understanding remains underexplored and lacks a standardized formulation** in current MLLMs, so we isolate it as a practical step to define and strengthen this capability. We view semantic and perceptual understanding as complementary, covering both **high-level** and **low-level** aspects of visual understanding.
>
> > **Q3/L2. Semantic capability after perceptual training**
> >
>
> As suggested by prior work, we believe that downstream training may lead to some change in pretrained capabilities. To examine semantic capability after perceptual training, we additionally evaluate **UniPercept** and its base model InternVL3-8B on MMBench [1], MMMU [2], and MMStar [3]. UniPercept shows a **mild drop** in general capability, but the magnitude is limited.
>
> |  | MMBench (en) | MMMU (val) | MMStar |
> | --- | --- | --- | --- |
> | InternVL3-8B | 86.00 | 58.67 | 68.40 |
> | UniPercept | 83.33 | 52.00 | 64.87 |
>
> UniPercept-Bench still requires substantial semantic understanding, as its QA pairs are semantically coherent and several perceptual categories rely on higher-level scene understanding beyond local appearance cues. Overall, this suggests that **perceptual** and **semantic** abilities are not independent, and perceptual-domain training does **not** cause severe semantic degradation, though a mild trade-off may exist. We will clarify this in the revision.
>
> [1] *MMBench: Is Your Multi-Modal Model an All-around Player?*, ECCV 2024, Yuan Liu *et al.*
>
> [2] *MMMU: A Massive Multi-discipline Multimodal Understanding and Reasoning Benchmark for Expert AGI*, CVPR 2024, Xiang Yue *et al.*
>
> [3] *Are We on the Right Way for Evaluating Large Vision-Language Models?*, NeurIPS 2024, Lin Chen *et al.*

---

> > ### Author Rebuttal · Reviewer_jUhM · 2026-04-03
> >
> > Thank you for the clarification. The authorsʼ response resolved my understanding on both the overall performance changes and the perspective from which the semantic and perceptual domains are framed. I also appreciate the authorsʼ note that the discussion on multi-domain perceptual integration (IAA + IQA) will be further clarified in the final version. Finally, the trade-off in semantic capability after perceptual training does not appear to be excessive, and thus seems to remain within an acceptable range. Overall, I would raise my score from 3 to 4.

---

> > > ### Author Response · Authors · 2026-04-03
> > >
> > > Thank you very much for your thoughtful follow-up and for taking the time to carefully reconsider our rebuttal. We sincerely appreciate your recognition that our clarifications helped resolve the concerns regarding both the overall performance changes and the framing of the semantic and perceptual domains.
> > >
> > > We are also grateful for your recognition that the semantic trade-off after perceptual training remains within an acceptable range. As noted, we will further clarify the discussion on multi-domain perceptual integration (IAA + IQA) in the final version.
> > >
> > > Thank you again for your constructive feedback and for raising your score from 3 to 4. We truly appreciate your support.

---

### Decision · Program_Chairs · 2026-04-30

**Decision:**

Accept (spotlight)

**Comment:**

The UniPercept addresses an important gap in perceptual-level image understanding by introducing a unified framework across Image Aesthetics Assessment, Image Quality Assessment, and the novel Image Structure & Texture Assessment domain. The benchmark is well-constructed with comprehensive experimental validation, and the baseline model trained via Domain-Adaptive Pre-Training and Task-Aligned Reinforcement Learning demonstrates strong performance with practical utility as a plug-and-play reward model for text-to-image generation. Reviewers appreciated the clear presentation, thorough experiments, and the commitment to open-source release of the dataset, model weights, and code. Initial concerns about the integration of VR and VQA tasks in the GRPO objective, the specific contribution of ISTA versus overall framework design, semantic capability preservation after perceptual training, and presentation issues were adequately addressed in rebuttal, with all four reviewers confirming their concerns were resolved and three raising their scores accordingly. While the paper has limitations including modest architectural novelty beyond the InternVL3-8B backbone and the ISTA score focusing more on richness than comprehensive structural quality, the work makes a substantive contribution by establishing the first unified perceptual-level understanding framework with strong experimental rigor and demonstrated practical impact. AC recommends accept.